

**Emission-dominated gas exchange of elemental mercury vapor over natural surfaces in China**
Xun Wang[1,2], Che-Jen Lin[1,3,4,*], Wei Yuan[1,2], Jonas Sommar [1], Wei Zhu [1], Xinbin Feng[1,*]
[1] State Key Laboratory of Environmental Geochemistry, Institute of Geochemistry, Chinese Academy of
Sciences, Guiyang, China
[2] University of Chinese Academy of Sciences, Beijing, China
[3] Center for Advances in Water and Air Quality, Lamar University, Beaumont, TX, USA
[4] Department of Civil and Environmental Engineering, Lamar University, Beaumont, TX, USA
**\* Corresponding Authors:**
Xinbin Feng                              Che-Jen Lin
Phone: +86-851-5895728          Phone: +1 409 880-8761
Fax + 86-851-5891609               Fax: +1 409 880-8121
Email: fengxinbin@vip.skleg.cn     E-mail: Jerry.Lin@lamar.edu
**Abstract**: Mercury (Hg) emission from natural surfaces plays an important role in global Hg cycling. The
present estimate of global natural emission has large uncertainty and remains unverified against field data,
particularly for terrestrial surfaces. In this study, a mechanistic model is developed for estimating the
emission of elemental mercury vapor ($Hg^0$) from natural surfaces in China. The development implements
recent advancements in the understanding of air-soil and air-foliage exchange of $Hg^0$ and redox chemistry
in soil and on surfaces, incorporates the effects of soil characteristics and landuse changes by agricultural
activities, and is examined through a systematic set of sensitivity simulations. Using meteorology simulated
by the Weather Research and Forecasting Model (WRF version 3.7), the exchange of $Hg^0$ between the
atmosphere and natural surfaces in Mainland China is estimated to be 465.1 Mg $yr^{-1}$, including 565.5 Mg
$yr^{-1}$ of emission from soils, 9.0 Mg $yr^{-1}$ of emission from water body, and 100.4 Mg $yr^{-1}$ uptake by vegetation.
The air-surface exchange is strongly dependent on the landuse and meteorology, with 9% of net emission
from forest ecosystems, 50% from shrubland, and savanna and grassland, 33% from cropland, and 8% from
other landuses. Given the large agricultural land area in China, farming activities play an important role on
the air-surface exchange in farmland. Particularly, rice field shift from a net sink (3.3 Mg uptake) during
April to October (rice planting) to a net source when the farmland is not flooded (November-March).
Summing up emissions from each landuse, more than half of the total emission occurs in summer (51%),
followed by spring (28%), autumn (13%) and winter (8%). Model verification is accomplished using
observational data of air-soil/air-water fluxes and Hg deposition through litterfall for forest ecosystems in
China and Monte Carlo simulations. In contrast to the earlier estimate by Shetty et al. (2008) that reported
large emission from vegetative surfaces using an evapotranspiration approach, the estimate in this study
shows natural emissions are primarily from grassland and dry cropland. Such an emission pattern may alter
the current understanding of Hg emission outflow from China as reported by Lin et al. (2010b) because of
a substantial natural Hg emission occurs in West China.



## 1. Introduction

Accurate inventories of mercury (Hg) emission is the foundation for assessing Hg global biogeochemical cycling (Selin, 2009;Streets et al., 2009;Streets et al., 2011). Hg emission from anthropogenic sources has been quantified and updated with reasonable consistency since the 1990s (Streets et al., 2009;Streets et al., 2011;Zhang et al., 2015;Zhang et al., 2016). In particular, the inclusion of the release from commercial products and modifications of Hg emission speciation profiles dependent on implementation and upgrade of air pollution control technologies have substantially reduced the uncertainty level of anthropogenic Hg emission estimates (Horowitz et al., 2014;Zhang et al., 2016). In contrast, estimates of natural Hg emission are poorly constrained within large uncertainties ($\pm 2000$ Mg yr$^{-1}$), limiting a comprehensive understanding of global/regional Hg cycling budgets (Pirrone et al., 2010;Wang et al., 2014b;Song et al., 2015). In light of the anticipated decrease of Hg emission from anthropogenic sources driven by the legally binding Minamata Convention (De Simone et al., 2015), a better quantification of natural Hg emission is critical in evaluating the effectiveness of policy actions (Selin, 2009;Pirrone et al., 2010;Song et al., 2015).

One of the challenges in predicting natural Hg emissions is to provide physical robust schemes for air-surface Hg$^0$ exchange. Such gas exchange schemes are complicated and involve not yet well understood processes in multiple environmental compartments, such as in terrestrial vegetative ecosystems. Thus, estimates from regression schemes obtained from correlations between Hg flux and environmental parameters (e.g., temperature, solar radiation, etc.) using limited datasets may not be representative (Xu et al., 1999;Bash et al., 2004;Lin et al., 2005;Gbor et al., 2006;Shetty et al., 2008;Selin et al., 2008;Smith-Downey et al., 2010) because relationship between measured fluxed and environmental parameters are based on limited field data that tend to be site-specific, which cannot account for the heterogeneity in soil properties and vegetative coverages. Although the bidirectional resistance schemes applied to describe the Hg$^0$ gas exchange may appropriate (Bash, 2010;Wang et al., 2014b;Wright and Zhang, 2015), they are limited by the availability of comprehensive soil property data and other physicochemical parameters such as Hg$^{II}$ reduction kinetics and physical characteristics of interfacial exchanges (Bash, 2010;Wang et al., 2014b), leading to inconsistences between simulated and measured fluxes. Other challenges including



acquiring and assimilating meteorology, landuse, and soil Hg and moisture content data into the model
scheme over a large geospatial coverage in China warrant for further model development to estimating
natural Hg emission (Wang et al., 2014b).
Advances in the fundamental understanding of $Hg^{II}$ reduction provide new opportunities to build a
more physically robust air-soil exchange scheme. These include constrained pseudo-first-order reduction
rate constant of $Hg^{II}$ in soil ($10^{-11}$ to $10^{-10}$ s$^{-1}$, (Scholtz et al., 2003;Qureshi et al., 2011) and in natural water
(0.2-1.0 h$^{-1}$, O'Driscoll et al., 2006;Qureshi et al., 2010), where the UV-band of actinic light is a primary
driver for $Hg^{II}$ photo-reduction in soils and water bodies rather than its visible part (Moore and Carpi,
2005;Si and Ariya, 2011). Besides constrained effective rate constants, the role of specific substructures
(e.g. functional groups such as -C(O)OH, -SH, -OH, ) of DOM in the reduction mechanism has been
pinpointed by kinetic studies of various model compounds(He et al., 2012;Si and Ariya, 2011, 2015). In
some of these specific reactions, the presence of molecular oxygen shows an inhibiting role in the reduction
to $Hg^0$ whereas e.g. the photo-reduction of HgII bound to R-SH compounds proceeds to $Hg^0$ irrespective of
dissolved $O_2$ (Si and Ariya, 2011). For the photo-reduction in dry soil, the first-order rate constants are
0.007-0.028 h$^{-1}$ for $HgCl_2$ coated over sand and 0.003-0.006 h$^{-1}$ for $Hg^{II}$ in a natural soil (Quinones and
Carpi, 2011). In the dark condition at room temperature (293 K), reduction rate constants of the reducible
Hg in soil are 0.001-0.002 h$^{-1}$ (Pannu, 2012).
The intercontinental transport of Hg from China has been considered to significantly contribute to Hg
deposition in North America (Jaffe et al., 2005;Strode et al., 2008;Lin et al., 2010b;Chen et al., 2014). Wang
et al. (2014a) suggested the Hg emissions from natural and anthropogenic sources were comparable. Using
an outdated model scheme, Shetty et al. (2008) estimated 462 Mg yr$^{-1}$ of Hg emitting into atmosphere in
China. This natural emission inventory has been used in East Asia CMAQ simulations to assess regional
Hg budgets (Lin et al., 2010b;Zhu et al., 2015a). However, the large uncertainty on the estimate of natural
Hg emission warrants a re-assessment of these earlier modeling efforts. In addition, the dataset of soil Hg
concentration in Shetty et al. (2008) is more than 3 decades old and requires updates to appropriately
represent spatially-resolved soil Hg contents that have been modified due to the rapid industrial
development and urbanization occurring in China since 1980s. In the meantime, the National Multi-Purpose



Regional Geochemical Survey (NMPRGS) of China has been completed in 2014 (Li et al., 2014). This
database provides soil Hg content for the agricultural and industrialized regions at resolution of 4 km, which
forms a basis of greatly reducing the uncertainty of Hg natural emission previously hampered by data
deficiency. In addition, a recent datasets of terrestrial flux in Mainland China reported in the literature
allows verification of model results and optimization of model schemes. In particular, recent field
measurements of $Hg^0$ air-surface exchange in China documented the flux characteristics over different land-
uses including urban-rural-remote differences and effects of crop rotation over agricultural lands(Fu et al.,
2008;Fu et al., 2010;Zhu et al., 2011;Fu et al., 2012;Fu et al., 2013a;Sommar et al., 2013a;Sommar et al.,
2013b;Zhu et al., 2013;Sommar et al., 2015a;Zhu et al., 2016).

104         In light of the scientific advancements and renewed data availability discussed above, this work

develops a state-of-the-science mechanistic model for estimating the natural Hg emission in China. For the
first time, the simulated natural emission flux is verified with field measurements over different land
surfaces in this modeling effort. The spatial, temporal and seasonal characteristics of the model-estimated
emissions over soil, vegetative surface and water are presented and compared to the estimates performed
by Shetty et al. (2008). The implications of the new natural emission estimate are discussed in terms of
chemical transport of Hg emission in China and the need for a re-assessment of mercury emission outflow
in China.

**2. Methods**
**2.1 Model description**
**2.1.1 Reduction of Hg$^{II}$ in soil**

116         Based on Hg$^{II}$ reduction mechanisms proposed in peer-reviewed literature(Moore and Carpi,

2005;Quinones and Carpi, 2011;Si and Ariya, 2011;Pannu, 2012), a new model scheme describing Hg$^0$
formation from Hg$^{II}$ reduction in soil is developed using three reaction pathways: (1) photo-reduction of
Hg$^{II}$ in soil pore water ($Hg_1^0$), (2) photo-reduction Hg$^{II}$ associated with soil particles ($Hg_2^0$), and (3) Hg$^{II}$
reduction through non-photochemical pathways ($Hg_3^0$). The production of Hg$^0$ in surface soil is calculated
as:



$\frac{dHg_1^0}{dt} = K_1 \cdot Hg_{s,m}$ (1)
$\frac{dHg_2^0}{dt} = K_2 \cdot Hg_{p,m}$ (2)
$\frac{dHg_3^0}{dt} = K_3 \cdot Hg_{t,m}$ (3)
where $K_1$ is the photo-reduction rate constant of Hg$^{II}$ in soil pore water (a comprehensive parameter list with
units is shown in Table 1), $K_2$ is the photo-reduction rate constant of Hg$^{II}$ associated with soil particles, $K_3$
is the Hg$^{II}$ reduction rate constant in soil through non-photochemical pathways, $Hg_{s,m}$ is the Hg$^{II}$ pool in soil
pore water, $Hg_{p,m}$ is the labile Hg$^{II}$ pool available for reduction in soil particles, $Hg_{t,m}$ is the total reducible
Hg$^{II}$ pool in soil. Based on the Arrhenius equation, $K_1$ and $K_2$ are parameterized as a function of solar
irradiance and soil temperature, and $K_3$ is parameterized as the function of soil temperature and soil moisture:
$K_1 = k_1 \cdot R \cdot \gamma \cdot e^{\frac{T-T_f}{T_f}\frac{E_a}{T}}$ (4)
$K_2 = k_2 \cdot R_i \cdot \gamma \cdot e^{\frac{T-T_f}{T_f}\frac{E_a}{T}}$ (5)
$K_3 = k_3 \cdot e^{\frac{T-T_f}{T_f}\frac{E_a}{T}} \cdot e^{\frac{\theta-\theta_f}{\theta_f}\frac{E_b}{\theta}}$ (6)
where $k_1$ and $k_2$ are the photo-reduction rate constants at the reference soil temperature ($T_f$, Table 1). $k_3$ is
the dark-reduction rate constant at the reference soil temperature and soil moisture ($\theta_f$, Table 1). $R$ and $R_i$
are total solar radiation in the soil profile and under the canopy, respectively. $\gamma$ is the ratio of UV over total
solar radiation. An empirical rule suggests that a 10°C temperature rise doubles reaction rates for many
chemical reactions near room temperature (Kissinger, 1957;Hood et al., 1975), which has been evinced to
apply for Hg$^{II}$ reduction in boreal soil (Pannu et al., 2014). In addition, Hg emission flux from soil substrate
doubles in the dark for a 25% increase of soil moisture content (Lin et al., 2010a). Based on these
observations, Equations 4-6 can be simplified as:
$K_1 = k_1 \cdot R \cdot \gamma \cdot 2^{\frac{T-T_f}{10}}$ (7)
$K_2 = k_2 \cdot R_i \cdot \gamma \cdot 2^{\frac{T-T_f}{10}}$ (8)
$K_3 = k_3 \cdot 2^{\frac{T-T_f}{10}} \cdot 2^{\frac{\theta-\theta_f}{25\%}}$ (9)
$R$ and $R_i$ in Equations 7-8 are calculated based on the Beer-Lambert law:



$R_i = R_0 \cdot e^{-K \cdot LAI}$                    (10)
$R = R_i \cdot \int_0^l e^{-k_r \cdot l} dl$                    (11)
where $R_0$ is solar irradiance above the canopy, $K$ is the canopy light extinction coefficient, $LAI$ is leaf area
index, $k_r$ is the light extinction coefficient in soil, $l$ is the depth of surface soil.
$Hg_{s,m}$ and $Hg_{p,m}$ and $Hg_{t,m}$ are calculated based on Lyon et al. (1997):
$Hg_{s,m} = \frac{[Hg_t] \cdot BD}{\theta + kd \cdot BD} \cdot V \cdot \theta \cdot 10^6$                    (12)
$Hg_{p,m} = \frac{[Hg_t] \cdot BD \cdot kd}{\theta + kd \cdot BD} \cdot BD \cdot V \cdot \varphi$                    (13)
$Hg_{t,m} = [Hg_t] \cdot BD \cdot V \cdot \varphi$                    (14)
where $[Hg_t]$ is the total Hg$^{\text{II}}$ concentration in soil, $BD$ is the soil bulk density, $\theta$ is the soil moisture, and
$V$ is the soil volume, $\varphi$ is the ratio of reducible Hg over total Hg in soil. $kd$ is the soil-water partition
coefficient and calculated following Lee et al. (2001) and Sauve et al. (2000):
$\log kd = r \cdot pH + s \cdot \log(1000 \cdot foc) + t$    (15)
where $f_{oc}$ is the fraction of organic carbon in surface soil. The values $r$; $s$ and $t$ are regression coefficients.
Following Obrist et al. (2014), we assume that the Hg$^0$ emission from soil is controlled by diffusion
process after Hg$^{\text{II}}$ reduction. Basing on the Fick's first law, the observed air-soil flux exchange can be
calculated as:
$F_{soil} = Hg_1^0 + Hg_2^0 + Hg_3^0 - D_{soil} \frac{GEM}{d/2} \Delta t$    (16)
$D_{soil} = 0.66 \cdot (\sigma - \theta) \cdot D_0$                    (17)
where $D_{soil}$ and $D_0$ are the Hg$^0$ vapor diffusion coefficient in soil and ambient air. GEM is the atmospheric
Hg$^0$ concentration, $\sigma$ is the soil porosity. Hence, during a time period $\Delta t$, the soil Hg$^0$ vapor compensation
point used in bidirectional resistance model can be derived as:
$\chi_g = \frac{(Hg_1^0 + Hg_2^0 + Hg_3^0) \cdot d/2}{D_{soil} \cdot \Delta t}$                    (18)

### 169    2.1.2 Updates for air-terrestrial exchanges

Extending from the two categories (vegetated surface canopy and bare land) considered in Wang et al.
(2014b), the terrestrial system is divided into four categories: vegetated surface with unsaturated soil





moisture (e.g., forest, grassland, shrubland, etc.), vegetated surface with saturated soil (i.e., rice paddy),
barren or sparsely vegetated land, and snow/ice surface. The governing equation for calculating $Hg^0$ air-
surface exchange over vegetated surfaces is:
$F_{cnp} = \frac{\Delta t}{(R_a + R_b)}(\chi_{cnp} - C_{atm})$          (19)
where $R_a$ is the aerodynamic resistance,   $R_b$ is the quasi-laminar sub-layer resistance, $C_{atm}$ is the
atmospheric Hg concentration, $\chi_{cnp}$ is the overall compensation point. For the canopy biomes with
unsaturated soil, $\chi_{cnp}$ is parameterized as Wang et al. (2014b):
$\chi_{cnp} = \frac{\frac{\chi_c}{R_c} + \frac{\chi_s}{R_s} + \frac{\chi_g}{R_g + R_{ac}} + \frac{C_{atm}}{R_a + R_b}}{\frac{1}{R_c} + \frac{1}{R_s} + \frac{1}{R_g + R_{ac}} + \frac{1}{R_a + R_b}}$          (20)
where $\chi_c$  is the cuticular compensation point,  $\chi_s$ is the stomatal compensation point, $R_c$  is the cuticular
resistance, $R_s$ is the stomatal resistance, $R_g$ is the soil diffusion resistance, $R_{ac}$ is the in-canopy
aerodynamic resistance (Table 1 in details). While for vegetated surface with saturate soil, $\chi_{cnp}$ is
parameterized as following:
$\chi_{cnp} = \frac{\frac{\chi_c}{R_c} + \frac{\chi_s}{R_s} + \frac{\chi_w}{R_w + R_{ac}} + \frac{C_{atm}}{R_a + R_b}}{\frac{1}{R_c} + \frac{1}{R_s} + \frac{1}{R_w + R_{ac}} + \frac{1}{R_a + R_b}}$          (21)
where $\chi_w$ is the soil compensation point,  $R_w$  is the water diffusion resistance.

186        The governing equation for air-surface exchange in barren or sparsely vegetated land, and snow/ice

surface is:
$F_{bls} = \frac{\Delta t}{R_a + R_b + R_g}(\chi_g - C_{atm})$          (22)
For bare land, $R_g$ is calculated following  Zhang and Lindberg (1999):
$R_g = \frac{d/2}{0.66 \cdot (\sigma - \theta) \cdot D_0}$          (23)
For snow/ice surface, $R_g$ is evaluated following  Zhang et al. (2012b):
$\frac{1}{R_g} = \frac{\alpha_{Hg^0}}{R_{g(SO_2)}} + \frac{\beta_{Hg^0}}{R_{g(O_3)}}$          (24)
where $R_{g(SO_2)}$  and $R_{g(O_3)}$  are the diffusion resistances of $SO_2$ and $O_3$, $\alpha_{Hg^0}$  is the $Hg^0$ scaling factor
based on $SO_2$, $\beta_{Hg^0}$  is $Hg^0$ scaling factor based on $O_3$. The formulation of $R_{g(SO_2)}$  and $R_{g(O_3)}$  has been
described previously (Zhang et al., 2003; Wang et al., 2014b). $\chi_g$  is assumed to be 3 ng m$^{-3}$ based on field



measurements at air-snow interface (Mann et al., 2015;Lalonde et al., 2003;Fain et al., 2007;Maxwell et al.,
2013). Other parameters in Equations 19-24 are described in details in Wang et al. (2014b).

**2.2 Model configuration and data**

The modeling domain is in Lambert Conformal projection, with 223×149 grid cells at 36-km spatial
resolution. The modeling period is one year (2013). Hourly meteorological data are prepared using the
Weather Research and Forecasting (WRF) model Version 3.7. Sensitivity analysis in Wang et al. (2014b)
showed that accurate model representation of environmental parameters (e.g., temperature, solar irradiance,
etc.) greatly improves the flux estimate. To obtain the best physics and dynamics options of WRF for the
China domain, a $L_{25}$ ($5^6$) orthogonal design of experiments is utilized (Supplementary material, Table **S1**).
The best combination of meteorological physics options is selected based on model evaluation metrics R
(correlation coefficient) and RMSE (root-mean-square error) between simulated outputs of each
combination of options and observed values in 750 meteorological stations. The selected physics options
are Thompson (Microphysics Options), Betts-Miller-Janjic (Cumulus Parameterization Options), RRTMG
(Radiation Physics Options) and BouLac (PBL Physics Options) based on the results of meteorological
model performance evaluation (Figure **S1**).
The datasets for surface soil properties (1 km spatial resolution) containing organic matters contents,
pH, bulk density and porosity are adopted from Shangguan et al. (2013). The land cover data (1 km spatial
resolution) is obtained from Ran et al. (2012). The land cover map represents the best available datasets
and follows the IGBP (International Geosphere-Biosphere Programme) classification system (Figure **S2**).
The ratio of rice planting fields in China during each month are classified following the method used in Liu
et al. (2013). The rice planting seasons are April to October in South China (including double rice planting),
and May to October in Northeast China (single rice planting). The LAI data, also with a 1 km spatial
resolution, are adopted from Yuan et al. (2011). The soil Hg content information utilized by Shetty et al.
(2008) is updated and greatly expanded with the comprehensive NMPRGS dataset (Li et al., 2014). These
high resolution datasets were re-gridded into the domain specification for each landuse using the spatial
tools in ArcGIS 10.1. The soil Hg content varies with landuse types, containing 119±9~211±83, 61±33 ~



197±96, 80±59~82±74, 80±59~82±74 and 31±15~162±83 of Hg for forest ecosystems, shrubland,
savanna/grassland, cropland, and other landuses, respectively (Table 1).
In the simulation, the $Hg^0$ concentration retrieved from the output of the Hg extension of Community
Multi-scale Air Quality modeling system (CMAQ-Hg) version 4.7 for the same modeling period is applied
to represent the ambient air concentration of $Hg^0$ (Lin et al., 2010b). The simulation does not incorporate
the feedback of the air-surface exchange to the air concentration because the feedback of the air-surface
exchange to the air concentration does not significantly modify the atmospheric $Hg^0$ concentration, and the
typical variation range of ambient $Hg^0$ concentration is not a sensitivity parameter for flux change (Wang
et al., 2014b). The model algorithms are coded in FORTRAN 90 and Network Common Data Form
(NetCDF) version 4.3. The gridded model results are visualized by the Visualization Environmental for
Rich Data Interpretation (VERDI) version 1.5.

**3 Results and discussion**
**3.1 Verification for soil Hg reduction scheme**
Values of all model parameters used in the simulation are showed in Table 1. The value of $k_1$ is assumed
to be $6\times10^{-9}$ $m^2$ $W^{-1}$ $s^{-1}$ based on the relationship between radiant intensity and apparent photo-reduction
rate constant in aerated solution observed by Si and Ariya (2011). Considering the 2 mm maximum
photolysis penetration depth in soil (Hebert and Miller, 1990), the measured rate constant in soil particles
(depth = 2.07 mm) from Quinones and Carpi (2011) is $2\times10^{-3}$ $m^2$ $W^{-1}$ $h^{-1}$ ($k_2$) with respect to the pool of
labile $Hg^{II}$ available for reduction. The value of $k_3$ is assumed to be $1.0\times10^{-3}$ $h^{-1}$ based on the average rate
constants for dark (thermal) reduction (Pannu, 2012). The mean ratio of reducible Hg in soil is assumed to
be 0.03 for the soil with vegetation based on measurements from Pannu (2012). No data is available for the
bare soil. Data from Lindberg et al. (1999) hints that Hg enriched desert soil (1400-5000 ng $g^{-1}$ total Hg )
produces a nominal $Hg^0$ efflux in the range from 40 to 60 ng $m^{-2}$ $h^{-1}$. Derived from back-calculation taking
pore diffusion into account, the fraction of reducible Hg is predicted at least 10 times lower ($\leq 0.003$) than
that in the soil with vegetation.





Sensitivity analyses using a box model for a typical forest soil are performed to gauge the selected rate
coefficients and the results are showed in Figure 2. With $k_1 = 6.0 \times 10^{-9}$ $m^2$ $W^{-1}$ $s^{-1}$, $k_2 = 2 \times 10^{-3}$ $m^2$ $W^{-1}$ $h^{-1}$, $k_3$
$= 1.0 \times 10^{-3}$ $h^{-1}$, the $Hg^0$ vapor concentration in soil porous media is estimated to be 4.5 ng $m^{-3}$, comparable
to the measured concentration ($4.1 \pm 2.0$ ng $m^{-3}$) in the surface forest floor (Moore and Castro, 2012),
suggesting that the selected values for the empirical constants appropriately represent environmental
condition. Generally, the range of $Hg^0$ vapor in all simulations is 1.5-6.7 ng $m^{-3}$. Less than 0.1% $Hg^0$ vapor
is from photo-reduction in soil solution as the Hg pool in soil solution is small ($\leqslant 0.1\%$ of total Hg
concentration). A ~16% fraction of the pore $Hg^0$ concentration derives from thermal $Hg^{II}$ reduction,
contributing to 0.5-1 ng $m^{-3}$ of $Hg^0$ vapor present in soil gas. $Hg^0$ soil gas concentrations are typically lower
than atmospheric Hg concentration (1-2 ng $m^{-3}$) in forest ecosystems (Carpi and Lindberg, 1998;Ericksen
and Gustin, 2006;Kuiken et al., 2008a;Kuiken et al., 2008b;Obrist et al., 2014;Fu et al., 2015) and suggest
forest floor acting as a sink at night. This is consistent with the sign of nocturnal fluxes observed over forest
floor settings (Carpi and Lindberg, 1998;Ericksen et al., 2006;Kuiken et al., 2008a;Kuiken et al., 2008b).
Moore and Carpi (2005) reported that the Hg flux under sun-lit condition is 3-5 times higher than the value
under dark condition. This developed model is capable of simulating such observation that the photo-
reduction in soil particles dominates the formation of $Hg^0$ vapor.
Figure 3 illustrates the model response to the model variables at the two experimental levels in Table
2. Noting that the two experimental levels represent the typical endpoints of environmental parameter
ranges and therefore the range of flux response to the possible variation of environmental factors can be
measured. On average, increasing soil bulk density from 0.1 to 1.5 g $cm^{-3}$, and Hg content from 50 to 1000
ng $g^{-1}$, and soil temperature from 0 to 30 ˚C, and solar radiation from 0 to 1000 W $m^{-2}$, will significantly
enhance the flux by 112-135 ng $m^{-2}$ $h^{-1}$. Additional 112-128 ng $m^{-2}$ $h^{-1}$ synergistic effects from the
combination of above parameters are also predicted. On the other hand, increasing leaf area index (LAI)
from 0 to 7 $m^2$ $m^{-2}$ reduces the flux by 131 ng $m^{-2}$ $h^{-1}$. In addition, LAI could offset the positive effects from
bulk density, soil Hg concentration, and solar radiation above canopy, leading to an additional -114 to -131
ng $m^{-2}$ $h^{-1}$ decrease, indicating that the canopy shading substantially constrains soil Hg evasion, consistent





with the shading could decrease 70-90% fluxes compared to non-shaded soils in filed measurements (Carpi
and Lindberg, 1998;Zhang et al., 2001;Choi and Holsen, 2009).
Compared to the earlier sensitivity analysis in Wang et al. (2014b), the soil organic matter content
appear not to considerably influence the simulated flux (p = 0.915). In the new scheme, the soil organic
matter is not incorporated into either $K_2$ or $K_3$, in accordance with the findings of Pannu (2012). While $Hg^0$
evasion from substrates coated with $HgCl_2$ and humic matter is inversely correlated with humic matter
content both in the dark and under irradiation, the inhibitory effect from humic matter is not linear to its
content (Mauclair et al. (2008). For instance, relatively small differences are observed at humic matter
content > 1% (Mauclair et al., 2008). Additionally, the effect of soil organic matter type has not been
comprehensively investigated (Zhang and Lindberg, 1999;Bash et al., 2007). Further study to quantify the
corresponding reduction rate constants associated with different types of soil organic matters (or species)
and solar radiation, as well as field flux data that relate the observed flux intensity to a given type of organic
matter, can further improve the present model parameterization.

## 288 3.2 Diurnal variation of natural $Hg^0$ emissions in China

Table 3 shows the annual mean air-surface fluxes for different landuse types. Annual mean air-foliage
fluxes range from -0.2 to -4.5 ng m$^{-2}$ h$^{-1}$, with the highest value over the woody savannas, and the lowest
over deciduous forests (Table 3). The diurnal variation for air-foliage flux is displayed in Figure 4. Higher
deposition occurs during early morning (8:00-10:00) and later afternoon (16:00-17:00) due to the suitable
air temperature and solar irradiance that induces Hg uptake by stomata. The rates of Hg uptake during
midday are comparatively weaker due to the stronger irradiance and higher temperature. This bimodal
pattern is consistent with field observations (Lindberg et al., 2002;Poissant et al., 2008;Fritsche et al.,
2008;Sommar et al., 2015a), suggesting that the model is capable of simulating the diurnal pattern of air-
foliage exchange of $Hg^0$. Such pattern in the modelling is caused by re-emission of the deposited Hg on the
surface foliage through photo-reduction under the strong solar radiation during noontime, and also the offset
effect from emissions from underlying soils. Except for urban lands, the strength of diurnal deposition for





the other landuse is controlled by LAI, solar radiation, and air temperature. The elevated atmospheric Hg
concentration is the important parameter to induce Hg uptake by growing foliage in urban lands.
Simulated mean air-soil fluxes range from 0.1 to 23.3 ng m$^{-2}$ h$^{-1}$, with the lowest flux over barren
vegetated lands and the highest over urban lands (Table 3). This suggests that the simulated air-soil fluxes
greatly vary over different landuses. There are distinct diurnal variations in terrestrial ecosystems (Figure
5). Such diurnal pattern is caused by the variation of solar radiation, close to zero at night and peaking at
13:00 to 15:00 (UTC+8). Similar diurnal patterns have been observed during filed measurements for forest,
grassland, and cropland in China (Feng et al., 2005;Fu et al., 2008;Fu et al., 2012;Zhu et al., 2013). The
degree of diurnal variability for each landuse in terrestrial ecosystems is highly related to the LAI. Higher
LAI gives a more intensive canopy shading and largely inhibits Hg evasion from soil under canopy. This is
also the main reason for relative weaker diurnal variation over forest soils compared to shrubland, grassland
and cropland (Figure 5). The synergistic interactions between low vegetation cover and high soil
concentration (Mean=162±83 ng g$^{-1}$) results in the strongest diurnal variation for urban land-types.
The simulated annual mean air-water flux is 3.4 ng m$^{-2}$ h$^{-1}$. The diurnal variability for air-water flux is
weaker since wind speed is a more influential driver than sun-light (Wang et al., 2014b), consistent with
the diurnal variation observed in field studies that meteorology and photochemical process are the primary
factors(Feng et al., 2002;Feng et al., 2003;Wang et al., 2006;Feng et al., 2008;Fu et al., 2010;Fu et al.,
2013a;Fu et al., 2013b).
Overall, the annual net natural emission in China is 465.1 Mg Hg (Table 3), including 565.5 Mg yr$^{-1}$
of emission from soil, 9.0 Mg yr$^{-1}$ of emission from water body, and 100.4 Mg yr$^{-1}$ deposition (uptake) on
vegetated landscapes. The annual quantity of emission from soil is comparable to the estimate (528 Mg yr$^{-1}$
$^{1}$) based on the scale-up calculation using measured air-soil fluxes (Fu et al., 2015a) that suggest emissions
from cropland and grassland are the most important contributor. Of the total Hg$^0$ emission estimated by the
model, 50% is from shrubland, savanna and grassland (C6-C11, 38% total landuse); 33% is from cropland
(C12-C13, 22% total landuse); 9% is from forest (C1-C5, 14% total landuse); and 8% is from other landuse
types. The forest contributes to 28% of Hg uptake by foliage; shrubland, savanna and grassland contribute
to 38%; cropland contributes to 33%; and other landuse types contribute to 1%.





Although soil Hg contents in forest ecosystems are 2-4 times higher than that in grassland and cropland,
total annual fluxes above the canopy (soil+foliage) of forest ecosystems are 1-6 times lower than the values
in other two types of landuses (Table 3). This highlights the importance of canopy cover in natural emission
process of $Hg^0$. It is noteworthy that the landuse data are based on the survey in 2000 (Ran et al., 2012).
During last 15 years, the forested area in China increased from 14.0% to 21.6% (FAO, 2014), benefiting
from implementation of governmental Grain for Green Project and stricter natural forest protection actions.
Assuming that annual mean air-surface fluxes are at the same level as in this study, the total quantity of
natural Hg emission in 2014 is approximately 5% smaller than this estimate because of the increasing forest
coverage. Given the forest coverage is projected to be 24% in 2030, and 26% in 2050 (FAO, 2014), the
total quantity of natural Hg emission in China during 2030-2050 would decrease 9-10 %.

**3.3 Spatial distribution of natural Hg emission in China**

The annual spatial distribution of air-foliage flux can be divided by the well-known geo-demographic
demarcation line, "Heihe-Tengchong Line" (Figure 6.1). The vegetation on the east side of the line is much
denser than on the west side of the line because of abundant annual precipitation ($\geqslant$ 800 mm, Figure **S2**),
which leads to much stronger $Hg^0$ uptake by vegetation (>90% of the grid cells have a flux below -1.0 ng
$m^{-2} h^{-1}$ on the east side, compared to >90% of the grid cells has a flux above -0.5 ng $m^{-2} h^{-1}$ on the west side).
There is an enhanced Hg deposition in South China (22˚N-27˚N,105˚E-113˚E, Figure 6.1) where fluxes
ranging -3.8 to -19.1 ng $m^{-2} h^{-1}$. This can be explained by an elevation in atmospheric Hg concentrations
and a more intense vegetative $Hg^0$ uptake. Field measurements suggest that this region has dense vegetation
(i.e., high LAI, Figure **S3**) and elevated (2-10 ng $m^{-3}$) atmospheric Hg concentration (Fu et al., 2015;Zhu,
2014), which enhances Hg uptake by foliage. Specifically, evergreen broadleaf forest has the highest LAI
compared to other type of forests (Liu et al., 2012) and shows enhanced Hg uptake (up to -4.5 ng $m^{-2} h^{-1}$
mean flux). Although the direct measurement of Hg deposition flux through vegetative uptake is still not
feasible presently, the measured Hg input through litterfall (Fu et al., 2015) suggested the rate of Hg uptake
by foliage could up to 4-12 ng $m^{-2} h^{-1}$, comparable to the simulation results in this study.



Figure 6.2 shows the spatial distribution of annual air-soil fluxes. There are three high flux regions (mean flux $\geqslant$ 10 ng m$^{-2}$ h$^{-1}$): cropland/grassland in South and Southwest China (mainly in Guangdong, Guangxi, Guizhou, Yunnan, Chongqing and Sichuan provinces), cropland in North China (Heibei, Henan, and Shangdong provinces), and grassland in North China (Inner Mongolia, Shanxi, and Shaanxi, and Xinjiang provinces). Such elevated fluxes in first two regions have been confirmed in field observations(Feng et al., 2005;Wang et al., 2005;Wang et al., 2006;Fu et al., 2008;Sommar et al., 2015b). Elevated fluxes in South and Southwest China are attributed to the elevated Hg concentration in soil (85% of grid cells has a soil Hg content $\geqslant$100 ng g-1, Figure 1). Interestingly, soil Hg content is not the primary factor causing the high fluxes in the other two regions (70% of grid cells has a soil Hg content $\leqslant$50 ng g$^{-1}$). Dry deposition of PBM and/or GOM plausibly supply to the reducible Hg in soil for gradual reduction and volatilization as Hg$^{0}$ (Sommar et al., 2015b). The relatively low LAI (Figure S3), strong solar irradiance and high soil temperature (Figure **S4-S5**) during summer/autumn contribute to the high simulated emissions. The lower simulated fluxes in desert regions compared to fluxes over grassland (Figure 6.2) are caused by the lower fraction of reducible Hg in soils.

Since the soil Hg$^{0}$ efflux is the primary source of natural Hg emission, the spatial distribution of the natural Hg emission is strongly influenced by air-soil flux (Figure 7.1). There is a distinct seasonal variation in the emission quantity: 8% in winter, 28% in spring, 51% in summer, and 13% in autumn (Figure 7.2-7.5). Elevated fluxes mainly cluster in South and Southwest China in winter because of higher soil Hg content (Figure 1), and relatively higher temperature and stronger irradiance. Highest correlation coefficients are found between the flux and soil Hg concentration and soil bulk density (Table 4), suggesting that the soil Hg$^{0}$ pool is a major factor influencing Hg emission in winter. From the cold to warm season, fluxes gradually increase from low latitude to high latitude with the seasonal change of temperature and solar radiation (Figure 7.2-7.4). Under the strong irradiance and temperatures during summer, >65% of the grid cells in the domain has a flux above 10 ng m$^{-2}$ h$^{-1}$ and the effect of soil Hg content becomes weaker (Table 4). In autumn, high flux occurs over the cropland of Central and North China, and over the regions with high soil Hg content (Figure 7.6) because of the decreasing temperature and solar irradiance. Overall,



72% of natural Hg emission occurs from May to September, with higher emission over grassland and
cropland in North China in these months.
It is worth noting that parts of regions in South China (23˚N-31˚N, 110˚E-120˚E, mainly in Fujian,
Jiangxi, Hunan, Hubei, and Anhui provinces) and Northeast China (39˚N-51˚N, 130˚E-134˚E, mainly in
Liaoning, Jilin and Heilongjiang provinces) have relatively lower fluxes (-6.9~9.0 ng m$^{-2}$ h$^{-1}$) during
summer and autumn time (Figure 7.4-7.5). In addition to the impact from the intensive canopy cover in
forests (Figure **S2**), the agricultural activities in these regions also contribute to the smaller fluxes. Based
on Liu et al. (2013), 60% croplands in these regions are flooded for rice planting in summer and autumn.
Field-scale flux measurement using micrometeorological methods (i.e., aerodynamic gradient method)
suggest that a typical oilseeds-rice rotated cropland in Southwest China is a significant source during
oilseeds planting seasons with fluxes of 10.1-89.4 ng m$^{-2}$ h$^{-1}$; and a mild sink during rice planting seasons
with fluxes of -3.4 to -15.8 ng m$^{-2}$ h$^{-1}$ (Zhu, 2014). The model also successfully simulates such a pattern,
with simulated fluxes at 1.1-101.5 ng m$^{-2}$ h$^{-1}$ (Figure 7.2-7.3) during winter and earlier spring when
croplands are not flooded and -3.5 to 1.5 ng m$^{-2}$ h$^{-1}$ during the rice growing season (Figure S6). Overall, 3.3
Mg Hg$^0$ is predicted to deposit into rice paddies during the rice growing season, with 56% of the deposition
occurring in summer, 41% in autumn, and 3% in late spring.

**3.4 Verification of model estimates**
For the first time, the simulated natural Hg emission in China is verified against field observational
data in this study. The dataset of Hg deposition through litterfall is utilized for verifying the simulated air-
foliage fluxes because of two reasons: (1) it has been shown that Hg deposition through litterfall dominates
dry deposition (≥70%) in forests of China (Fu et al., 2015); and the annual Hg deposition through litterfall
has been used as a surrogate to constrain air-foliage fluxes in forest ecosystems (Risch et al., 2012;Zhang
et al., 2012a), and (2) the litterfall data in China have are more comprehensive for different forest types
compared to the air-foliage flux measured by enclosure methods. For verifying the exchange fluxes over
water and soil surfaces, the flux measurements over forest soil, grassland, cropland and water body in China
(Table **S2**) are utilized.





To estimate the annual Hg deposition through litterfall in the study domain, Monte Carlo simulation
(described in details in SI) is applied for constructing the probability distribution of the litterfall deposition
based on litter biomass production and litterfall Hg concentration in China reported in peer-reviewed
literature (Figure 8). The sampling locations include 20 sites in Tibetan Plateau, 27 sites for evergreen
forests, and 12 sites for deciduous forests. The quality-assured data of litter biomass production (number of
replicates $\geqslant$ 3, collector size = 1 m$^2$) are obtained from the China National Knowledge Infrastructure
(CNKI). This dataset includes the measurements at 5 sites in Tibetan Plateau, 277 sites for evergreen forests,
74 sites for deciduous forests, and 61 sites for mixed forests.
Figure 8 shows the dataset of Hg concentration in litterfall. The Hg concentration for evergreen forest
ranges from 17 to 120 ng g$^{-1}$ with a mean of 52±26 ng g$^{-1}$. For deciduous forest, the range is 21-62 ng g$^{-1}$
with a mean of 38±12 ng g$^{-1}$. The difference between the concentration observed in evergreen forests and
in deciduous forests is significant (paired $t$ test, p < 0.05). The Hg concentration in litters for deciduous
forest in China is compared to the values reported for the same forest type in Europe and North America (p
= 0.101). Hg deposition through litterfall in evergreen broadleaf forest (C2) range from 26 to 72 μg m$^{-2}$ y$^{-1}$
(n=5 sites) with a mean of 43±27 μg m$^{-2}$ y$^{-1}$ (Fu et al., 2015;Ma et al., 2015;Wang et al., 2009), consistent
with the Hg deposition estimated by Monte Carlo simulation (mean=37±19 μg m$^{-2}$ y$^{-1}$; 95% confidence
interval is 4-89 μg m$^{-2}$ y$^{-1}$). The model-estimated Hg deposition for C1, C3, C4, and C5 is 22±10, 15±7,
16±11, and 17±8 μg m$^{-2}$ y$^{-1}$, respectively.
The measured air-soil flux (Table **S2**) ranges from -1.4 to 20.7 ng m$^{-2}$ h$^{-1}$ over forest soil (n=19; mean=
6.1±5.1 ng m$^{-2}$ h$^{-1}$), from -18.7 to 114 ng m$^{-2}$ h$^{-1}$ over grassland (n=14; mean= 26±36 ng m$^{-2}$ h$^{-1}$), from -4.1
to 135 ng m$^{-2}$ h$^{-1}$ over cropland (n=33; mean= 21.3±36.7 ng m$^{-2}$ h$^{-1}$). For water body (n=51), the flux range
is 0-43.8 ng m$^{-2}$ h$^{-1}$ with a mean of 4.6±6.6 ng m$^{-2}$ h$^{-1}$ (Table **S2**). The mean flux in the warm season (May
to October) is substantially higher than those in the cold seasons: 3.3 times for water surface (p=0.004), 3.2
times higher for forest soil (p=0.08), and 1.4 times for cropland (p=0.50). A reverse trend is found for
grassland, which has higher mean flux in cold seasons (50% higher, p=0.36).
Figure 9.1 compares the model estimate and the mean and uncertainty level estimated by Monte Carlo
simulation using field data. The annual Hg uptake simulated by the bidirectional exchange model is not



significantly different from the field observations (p>0.05, t-test), demonstrating the model capability for
simulating the air-foliage flux. Figure 9.2 shows the scatter plot of the measured versus model predicted
flux over soil and water ($R^2$=0.73). Modeling results for soil/water surfaces and over soil under forest
canopy also agree with filed measurements (Figure 9.3-9.4). The simulation somewhat underestimates the
high fluxes ($\geqslant$ 30 ng m$^{-2}$ h$^{-1}$, Figure 9.2) measured over grassland and cropland (Figure 9.4).
The underestimated fluxes over grassland and cropland can be attributed to several possible reasons.
One is the bias caused by the comparatively coarser spatial resolution (36 km) of meteorological parameters
and soil properties that limit the reproduction of the instantaneously measured fluxes at observational sites.
In addition, the kinetic constants observed by laboratory study are used in the simulation for grassland and
cropland and potentially introduce uncertainty. Further studies focusing on in-situ measurement of Hg$^{II}$
reduction rate in soil can help constrain the model simulation result. Furthermore, limited mechanistic
understanding on Hg dry deposition (Gustin et al., 2015) and on the fate of deposited Hg (Lindberg et al.,
2007;Gustin et al., 2008b;Gustin et al., 2015;Ariya et al., 2015) complicates quantifying the contribution
of dry deposition to Hg emission from soil. Finally, the uncertainties caused by flux quantification
methodology (Lin et al., 2012;Zhu et al., 2015b, c) and the typically short transient campaign periods
(mostly ranging from several days to a couple of weeks) can also be a factor(Feng et al., 2005;Fu et al.,
2008;Fu et al., 2012;Fu et al., 2015;Zhu et al., 2015b, c). Improvement on flux methods and extended
campaign periods at more study sites for cropland/grassland will improve the model estimates.

**3.5 Comparison with earlier estimates and implications on Hg emission outflow in China**

Figures **S7** and **S8** show the gridded natural Hg emission in the East Asian Domain reported by Shetty
et al. (2008) and Wang et al. (2014a), which have two distinct differences compared to the new model
estimate in this study. One is regarding the role of vegetation in natural Hg emission, the other is spatial
distribution of the emission. Vegetation is clearly assigned as a substantial sink of Hg$^0$ based on the
mechanistic model algorithms in this study; and the shading of vegetation suppress Hg evasion from soil
under canopy. In contrast, vegetation is considered a major source, accounting for 76% total emissions in
Shetty et al. (2008) because earlier models treat Hg evasion similar to the evapotranspiration process that





transport Hg from root zone through vascular tissues in foliage (Gbor et al., 2007;Shetty et al., 2008).
However, recent experimental evidence using stable Hg isotope tracers points to exclusion of this pathway
for cereal plants (Cui et al., 2014). In addition, Hg isotopic signatures in air and foliage samples (Demers
et al., 2013;Yin et al., 2013) and during air-foliage exchange process (Graydon et al., 2006;Gustin et al.,
2008a) indicate uptake of atmospheric Hg by foliage, pointing to vegetation as a $Hg^0$ sink. Also in contrast
to the spatial distribution of the emission in this study, earlier $Hg^0$ emission estimates occur mainly in the
regions on the east side of the "Heihe-Tengchong Line" (Shetty et al., 2008;Wang et al., 2014a). Such spatial
distribution in Shetty et al. (2008) is caused by spatial distribution of vegetation as the vegetation is the
most important contributor for Hg emission, and in Wang et al. (2014a) is caused by spatial distribution of
soil Hg concentration as soil Hg concentration shapes natural Hg emission in simple regression schemes.
Furthermore, this study advances upon the earlier estimates (Shetty et al., 2008;Wang et al., 2014a) in
two fronts. Firstly, the recent soil survey data including soil Hg content and other soil characteristics is a
major advantage in this study. The soil Hg data applied in Shetty et al. (2008) is outdated with a coarse
spatial resolution; while the data in Wang et al. (2014a) is based on the output of the global terrestrial Hg
model in GEOS-Chem, calculated from Hg/C ratios. In addition, the mechanistic model scheme that
comprehensively treats the meteorological parameters and physicochemical properties of soil represents the
air-surface exchange more appropriately compared to the earlier, undeveloped regression schemes (Shetty
et al., 2008;Wang et al., 2014a). With the model results verified against the field flux measurements, the
natural emission quantity and spatial distribution in this study represent the state of science estimate of air-
surface exchange of $Hg^0$ in China.
Although the total quantity of annual natural emission estimated by the new model developed in this
study is comparable to in earlier estimates (400-600 Mg yr$^{-1}$) by Shetty et al. (2008) and Wang et al. (2014a),
the distinct spatial distribution of natural emissions simulated in this study may alter the current
understanding of Hg emission outflow from China as reported by Lin et al. (2010b). The outflow of Hg
emissions in China is mainly driven by the prevailing west-wind drift (Lin et al., 2010b;Chen et al., 2014).
The larger natural Hg emission in the west side of model domain results in a longer residence time of evaded
Hg, which can be more readily oxidized and then deposited within the domain. Furthermore, the dense





vegetation in the east side of the domain also enhance the uptake of atmospheric $Hg^0$. Such a spatial pattern
may lead to substantially larger domestic deposition and smaller quantity of outflow compared to the modes
estimates by Lin et al. (2010). We are presently to reassess the emission outflow using a regional chemical
transport model (e.g., CMAQ-Hg) and a similar mass balance approach by Lin et al. (2010); and will report
the model results in a future paper.

**4. Conclusions**
Using a mechanistic model incorporating the present state of understanding in Hg transformation in
soils and on foliage surface with up-to-date datasets of soil characteristics and landuse changes, the natural
emission of elemental mercury vapor in China is estimated to be 465.1 Mg yr$^{-1}$, including 565.5 Mg yr$^{-1}$ of
emission from soils, 9.0 Mg yr$^{-1}$ of emission from water bodies, and -100.4 Mg yr$^{-1}$ deposition (uptake) by
vegetation. The air-surface exchange is strongly dependent on landuse and meteorology, with 9% of net
emission from forest ecosystems, 50% from shrubland, savanna and grassland, 33% from cropland, and 8%
from other landuses. Given the large agricultural land area in China, farming activities play an important
role on the air-surface exchange. Particularly, rice fields shift from a net sink (3.3 Mg uptake) during the
growing season in rice paddy to a net source during the season when the farmland is not flooded. The
estimated natural $Hg^0$ emission in this study yields similar $Hg^0$ evasion quantity but exhibits contrasting
spatial distribution compared to the estimate by Shetty et al. (2008). The difference in the spatial emission
patterns may alter the current understanding of Hg emission outflow from China as reported by Lin et al.
(2010b) because of a substantial amount of natural $Hg^0$ emission occurs in West China.
For future model improvement, studies focusing on fundamental understanding of $Hg^{II}$ reduction in
soil (especially the role of soil organic matter, contribution of photochemical and non-chemical pathways,
and radiation transfer in soil) and air-vegetation exchange mechanisms are needed. Continuous
improvement on the data quality of soil characteristics and Hg content is also essential. Availability of field
data for modeling performance evaluation is also important for constraining the model. In particular, data
of air-foliage flux, and air-soil flux over cropland and grassland in the remote regions of North China is
also valuable for model calibration.



**Acknowledgements**
This work was funded by National "973" Program of China (2013CB430003), and State Key Laboratory
of Environmental Geochemistry, IGCAS. The funding support is gratefully acknowledged.

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



Table 1 Model variables, constants and rate coefficients used in the model simulation.

| Term | Description | Values/units | References/Sources |
|---|---|---|---|
| $Hg_1^0$ | $Hg^0$ formed by photo-reduction in solution | ng m$^{-2}$ h$^{-1}$ | |
| $Hg_2^0$ | $Hg^0$ formed by photo-reduction in particles | ng m$^{-2}$ h$^{-1}$ | |
| $Hg_3^0$ | $Hg^0$ formed by the dark-reduction in soil | ng m$^{-2}$ h$^{-1}$ | |
| $K_1$ | Photo-reduction rate in soil solution | s$^{-1}$ | |
| $K_2$ | Photo-reduction rate constant in particles | s$^{-1}$ | |
| $K_3$ | Dark-reduction rate constant in soil | s$^{-1}$ | |
| $Hg_{s,m}$ | $Hg^{II}$ pool in soil pore water | ng m$^{-2}$ | |
| $Hg_{p,m}$ | Reducible $Hg^{II}$ pool in soil particles | ng m$^{-2}$ | |
| $Hg_{t,m}$ | Total reducible $Hg^{II}$ pool in soil | ng m$^{-2}$ | |
| $T$ | Temperature | K | WRF |
| $\theta$ | Soil moisture | % | WRF |
| $R$ | Total solar radiation | W m$^{-2}$ | WRF |
| $BD$ | Soil bulk density | g cm$^{-3}$ | (Shangguan et al., 2013) |
| $LAI$ | Leaf area index | m$^2$ m$^{-2}$ | WRF (Yuan et al., 2011) |
| $[Hg_t]$ | Total $Hg^{II}$ concentration in soil | ng g$^{-1}$ | |
| $V$ | Soil volume | m$^3$ | |
| $kd$ | soil-water partition coefficient | L kg$^{-1}$ | |
| GEM | Atmospheric $Hg^0$ concentration | ng m$^{-3}$ | (Lin et al., 2010b) |
| $\sigma$ | Soil porosity | % | (Shangguan et al., 2013) |
| $D_{soil}$ | $Hg^0$ vapor diffusion coefficient in soil | m$^2$ s$^{-1}$ | |
| $D_0$ | $Hg^0$ vapor diffusion coefficient in ambient air | 1.31×10$^{-5}$ m$^2$ s$^{-1}$ | (Scholtz et al., 2003) |
| $F_{cnp}$ | The flux over canopy biomes | ng m$^{-2}$ h$^{-1}$ | |
| $\Delta t$ | Time duration | s | |
| $R_a$ | Aerodynamic resistance | s m$^{-1}$ | (Wang et al., 2014b;Zhang et al., 2012b) |
| $R_b$ | Quasi-laminar sub-layer resistance | s m$^{-1}$ | Wang et al., 2014b;Zhang et al., 2012b |
| $C_{atm}$ | Atmospheric Hg concentration | ng m$^{-3}$ | Wang et al., 2014b;Zhang et al., 2012b |
| $\chi_{cnp}$ | The total compensation point | ng m$^{-3}$ | Wang et al., 2014b;Zhang et al., 2012b |
| $\chi_c$ | Cuticular interfaces compensation point | ng m$^{-3}$ | Wang et al., 2014b;Zhang et al., 2012b |
| $\chi_s$ | Stomatal interfaces compensation point | ng m$^{-3}$ | Wang et al., 2014b;Zhang et al., 2012b |
| $\chi_g$ | Soil interfaces compensation point | ng m$^{-3}$ | Wang et al., 2014b;Zhang et al., 2012b |
| $R_c$ | cuticular resistance | s m$^{-1}$ | Wang et al., 2014b;Zhang et al., 2012b |
| $R_s$ | stomatal resistance | s m$^{-1}$ | Wang et al., 2014b;Zhang et al., 2012b |
| $R_g$ | soil diffusion resistance | s m$^{-1}$ | Wang et al., 2014b;Zhang et al., 2012b |
| $R_w$ | water diffusion resistance | s m$^{-1}$ | Wang et al., 2014b;Zhang et al., 2012b |
| $R_{g(SO_2)}$ | SO$_2$ soil diffusion resistance | s m$^{-1}$ | Wang et al., 2014b;Zhang et al., 2012b |
| $R_{g(O_3)}$ | O$_3$ soil diffusion resistance | s m$^{-1}$ | Wang et al., 2014b;Zhang et al., 2012b |
| $d$ | Light penetration into soil column | 2 mm | (Hebert and Miller, 1990) |



| | | | |
|---|---|---|---|
| $\gamma$ | Ratio of UV radiation over total radiation | 0.08 | (Moan, 2001) |
| $K$ | Canopy light extinction coefficient | 0.56 | (Zhang et al., 2014) |
| $k_r$ | Light extinction coefficient in soil | 3 mm$^{-1}$ | (Ciani et al., 2005) |
| $k_1$ | Photo-reduction rate constant in soil solution | 6×10$^{-9}$ m$^2$ W$^{-1}$ s$^{-1}$ | (Si and Ariya, 2011) |
| $k_2$ | Photo-reduction rate constant in soil particles | 2×10$^{-3}$ m$^2$ W$^{-1}$ h$^{-1}$ | (Quinones and Carpi, 2011) |
| $k_3$ | Dark-reduction rate constant in soil | 1.0×10$^{-3}$ h$^{-1}$ | (Pannu, 2012) |
| $T_f$ | Reference soil temperature | 32℃(Eq.8), 20℃ (Eq. 7,9) | (Pannu, 2012;Quinones and Carpi, 2011) |
| $\theta_f$ | Reference soil moisture | 25% | (Lin et al., 2010a) |
| $r$ | Empirical value from regression | 0.52 | (Lee et al., 2001;Sauve et al., 2000) |
| $s$ | Empirical value from regression | 0.89 | (Lee et al., 2001;Sauve et al., 2000) |
| $t$ | Empirical value from regression | -0.71 | (Lee et al., 2001;Sauve et al., 2000) |
| $\varphi$ | Ratio of reducible Hg in soil | 0.003 (bare), 0.03(others) | (Pannu, 2012) |
| $\alpha_{Hg^0}$ | Scaling factor of reactivity Hg | 0 | (Wang et al., 2014b) |
| $\beta_{Hg^0}$ | Scaling factor of reactivity Hg | 0.1 | (Wang et al., 2014b) |
| $Hg_w^{2+}$ | Hg$^{II}$ concentration on leaf | 3 ng m$^{-2}$ leaf | (Laacouri et al., 2013) |





Table 2 Examined model variables and the experimental levels of factorial design for air-soil exchange.

| Terms | Description | Low level | High level |
|---|---|---|---|
| BD | Soil bulk density (g cm$^{-3}$) | 0.1 | 1.5 |
| pH | Soil pH (dimensionless) | 4 | 9 |
| P | Soil total porosity (%) | 0.05 | 0.5 |
| T | Soil temperature ($^{o}$C) | 0 | 35 |
| SMOIS | Soil moisture (%) | 0.05 | 0.5 |
| R0 | Solar irradiance above canopy (w m$^{-2}$) | 0 | 1000 |
| LAI | Leaf area index (dimensionless) | 0 | 7 |
| GEM | Atmospheric Hg0 concentration (ng m$^{-3}$) | 1.5 | 5 |
| Hgs | Hg concentration in soil (ng g$^{-1}$) | 10 | 400 |
| Foc | Soil organic matter content (%) | 0.5 | 30 |
| k1 | photo-reduction rates in soil solution (m$^2$ W$^{-1}$ s$^{-1}$) | $3 \times 10^{-9}$ | $9 \times 10^{-9}$ |
| k2 | photo-reduction rates in soil particles (m$^2$ W$^{-1}$ h$^{-1}$) | $0.7 \times 10^{-3}$ | $3.0 \times 10^{-3}$ |
| k3 | Non-photo-reduction rates (thermal, h$^{-1}$) | $1.0 \times 10^{-3}$ | $2.3 \times 10^{-3}$ |




Table 3. Mean annual air-surface fluxes, and annual total Hg emissions from individual landuse. SHg is the
Hg content in surface soil (0-10 cm), FF is the $Hg^0$ flux over foliage, and FS is the $Hg^0$ flux over soil.

| Type | Description | Area(%) | SHg (ng g$^{-1}$) | FF (ng m$^{-2}$ h$^{-1}$) | Leaf(Mg) | FS(ng m$^{-2}$ h$^{-1}$) | Soil(Mg) | Tot(Mg) |
|---|---|---|---|---|---|---|---|---|
| C1 | Evergreen needleleaf forest | 5.7 | 186±74 | -2.8 | -13.5 | 6.9 | 35.2 | 21.7 |
| C2 | Evergreen broadleaf forest | 2.6 | 184±35 | -2.6 | -6.5 | 6.2 | 16.1 | 9.5 |
| C3 | Deciduous needleleaf forest | 0.1 | 119±9 | -0.2 | -0.03 | 0.7 | 0.1 | 0.1 |
| C4 | Deciduous broadleaf forest | 3.3 | 143±47 | -1.2 | -3.7 | 2.7 | 8.3 | 4.6 |
| C5 | Mixed forest | 2.4 | 211±83 | -2.2 | -4.5 | 4.7 | 10.5 | 6.0 |
| C6 | Closed shrubland | 5.2 | 115±77 | -3.2 | -14.1 | 5.6 | 26.0 | 11.9 |
| C7 | Open shrubland | 0.6 | 155±72 | -1.4 | -0.7 | 10.8 | 6.5 | 5.7 |
| C8 | Woody savanna | 0.3 | 197±96 | -4.5 | -1.0 | 12.9 | 3.2 | 2.2 |
| C9 | Savanna | 0.0 | 157±47 | -0.6 | -0.003 | 0.1 | 0.0 | 0.0 |
| C10 | Grassland | 31.8 | 61±33 | -0.8 | -20.3 | 8.0 | 221.8 | 201.4 |
| C11 | Permanent wetland | 1.1 | 74±24 | -0.8 | -0.8 | 9.8 | 10.0 | 9.2 |
| C12 | Cropland | 20.5 | 80±59 | -1.8 | -31.6 | 10.0 | 179.0 | 147.5 |
| C13 | Cropland mosaic | 1.6 | 82±74 | -2.0 | -3.0 | 6.7 | 10.1 | 7.2 |
| C14 | Urban land | 0.2 | 162±83 | -3.6 | -0.7 | 23.3 | 4.4 | 3.7 |
| C15 | Snow and ice | 0.8 | 31±15 | | | 2.0 | 3.1 | 3.1 |
| C16 | Barren vegetated land | 21.6 | 35±7 | | | 1.5 | 22.2 | 22.2 |
| C17 | Bodies of water | 2.2 | | | | 3.4 | 9.0 | 9.0 |
| Sum | | 100.0 | | | -100.4 | | 565.5 | 465.1 |




Table 4. Pearson correlations between mean total fluxes and major controlling environmental parameters
in each season. "**" means $p < 0.01$ and "*" means $p < 0.05$.

| Term | Winter | Spring | Summer | Autumn |
|---|---|---|---|---|
| LAI | -0.14* | -0.24** | -0.39** | -0.30** |
| Soil temperature | 0.27** | 0.35** | 0.54** | 0.38** |
| Solar radiation | 0.27** | 0.32** | 0.59** | 0.36** |
| Soil Hg concentration | 0.47** | 0.13* | 0.02 | 0.39** |
| Soil bulk density | 0.41** | 0.16* | 0.04 | 0.32** |



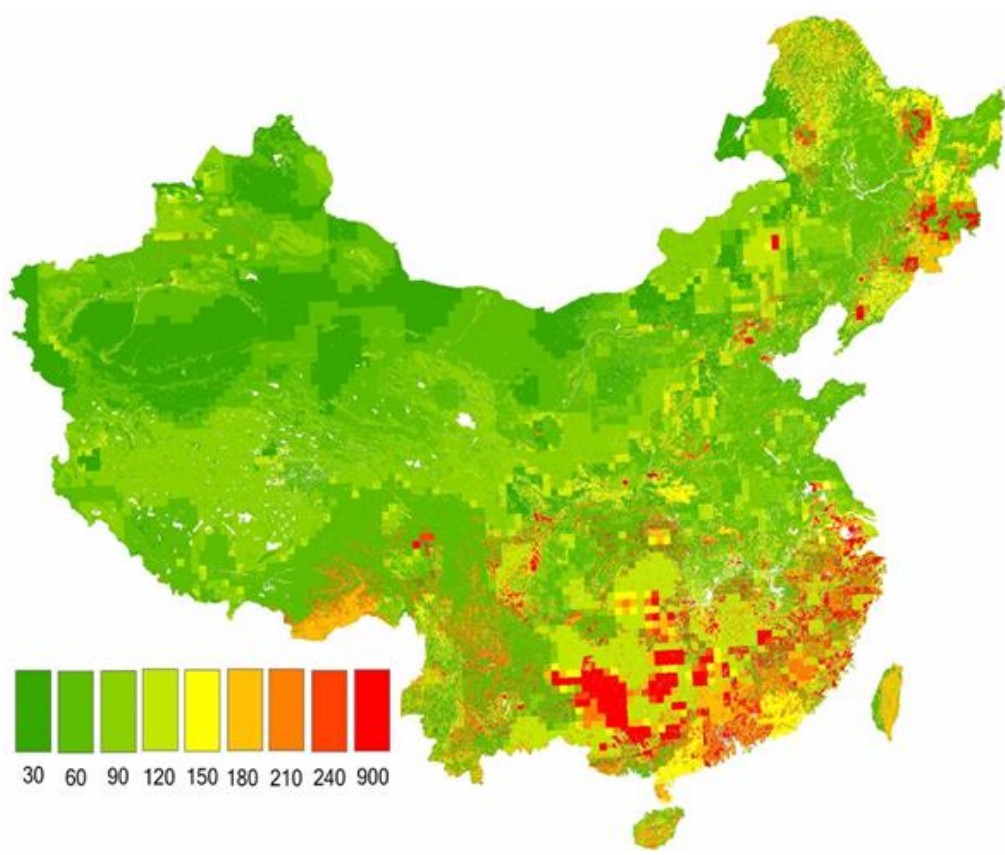


Figure 1. Updated Hg concentrations (ng g$^{-1}$) in surface soil of China. Sampling areas in NMPRGS covers
most agriculturally and industrially developed regions of eastern and central China, and is presented in
more details in Li et al. (2014).





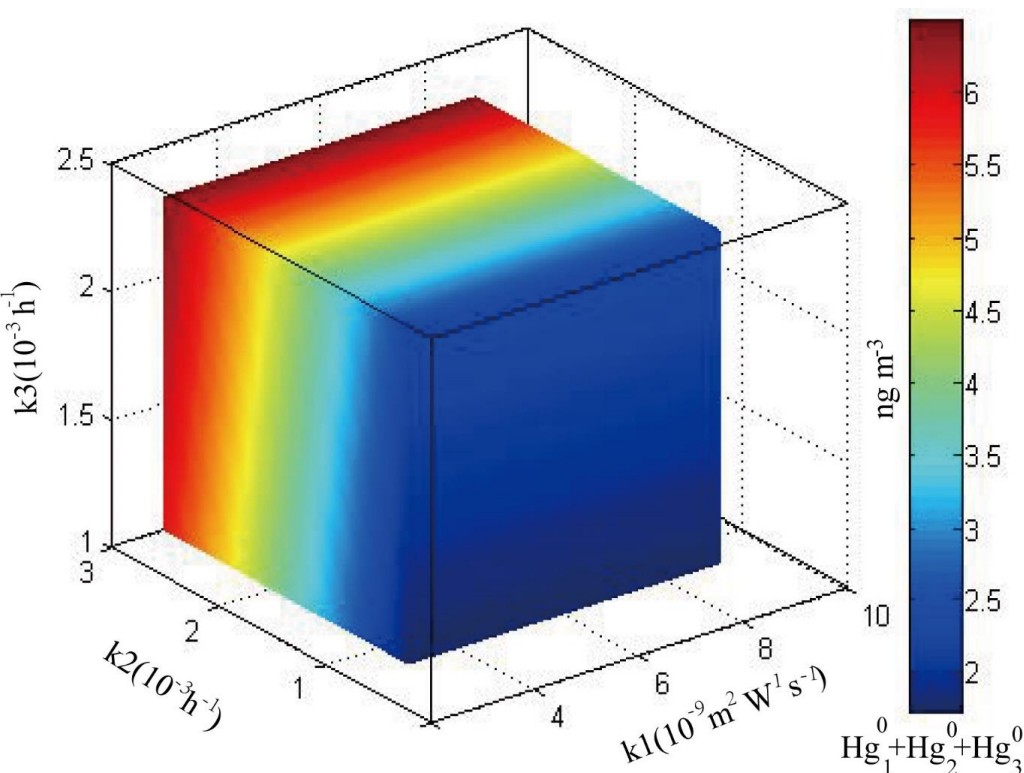


Figure 2. Variation of $Hg^0$ concentrations in the forest soil pore gas using the typical ranges of kinetic constants for $Hg^{II}$ reduction in soil (see text for details): $k_1 = 3.0\text{-}9.0\times10^{-9}$ $m^2$ $W^{-1}$ $s^{-1}$, $k_2 = 0.7\text{-}3.0\times10^{-3}$ $m^2$ $W^{-1}$ $h^{-1}$, $k_3 = 1.0\text{-}2.3\times10^{-3}$ $h^{-1}$, soil Hg content=150 ng $g^{-1}$, pH=5, soil organic content = 20%, soil bulk density = 0.7 g $m^{-3}$, solar irradiance = 1000 W $m^{-2}$, soil temperature = 25 °C, LAI = 5 $m^2$ $m^{-2}$, soil moisture content = 20%, and soil soil porosity = 40%.






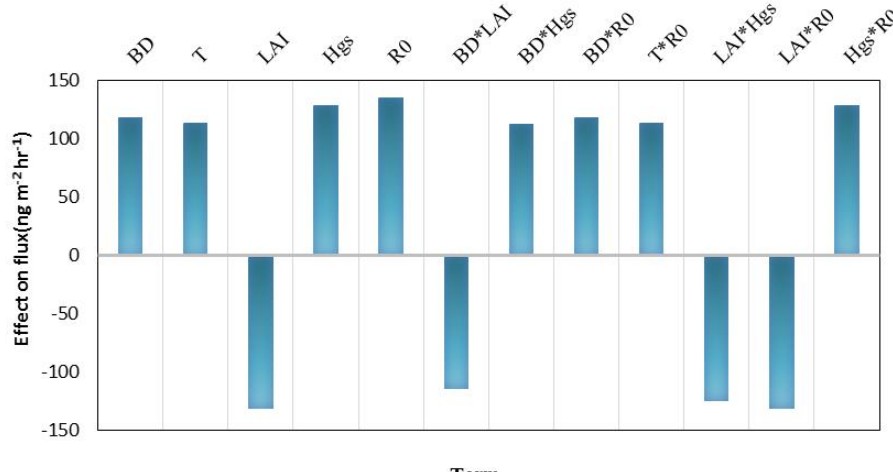



Figure 3. Sensitivity analysis on model parameters for air-soil exchange using a 2-level factorial design after pre-screening the model variables shown in Table 2 for the identified significant factors. The effects shown in the figure is based on a significance level of 95% (i.e., $p<0.05$).




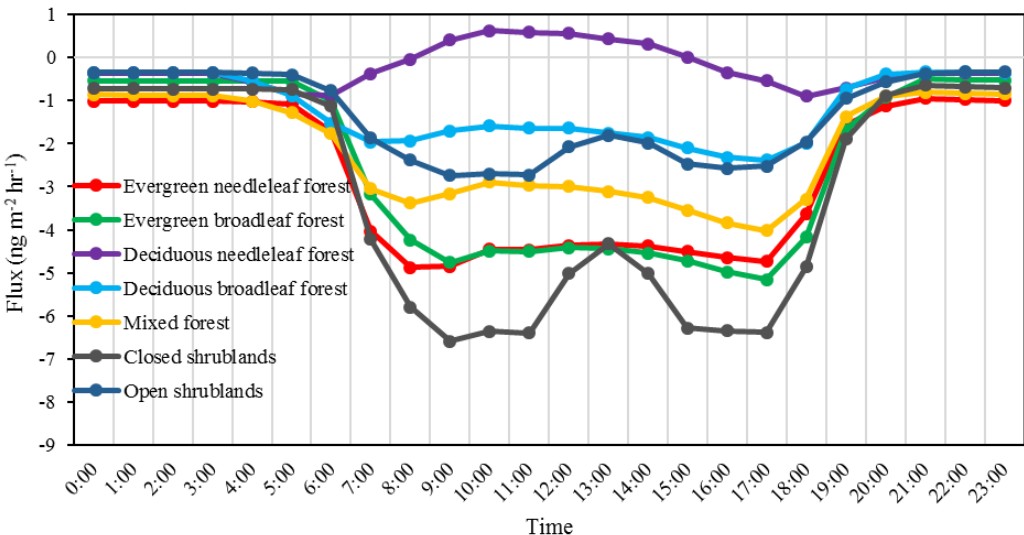


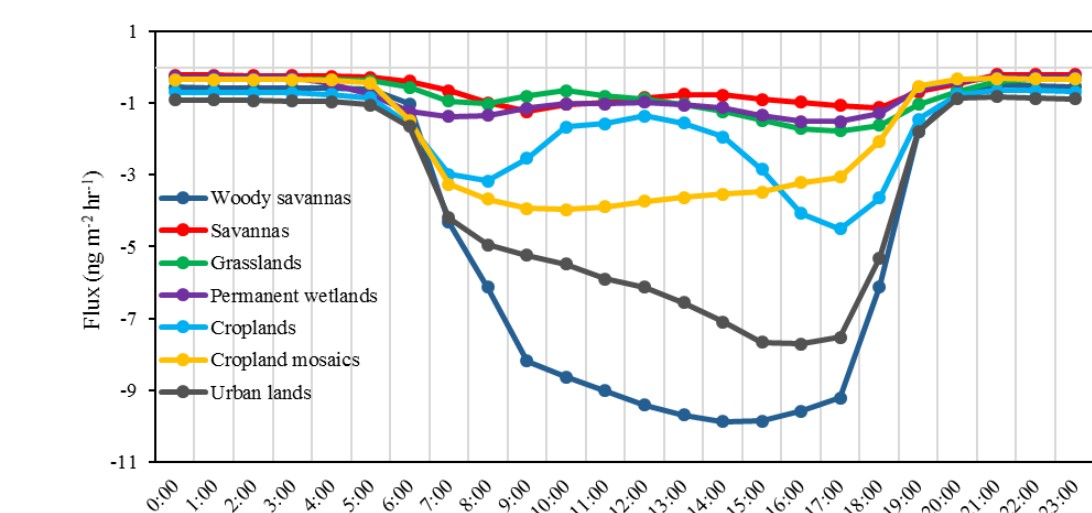


Figure 4. Diurnal variation of mean simulated exchange fluxes of Hg$^0$ over canopy in the model domain
(UTC+8).





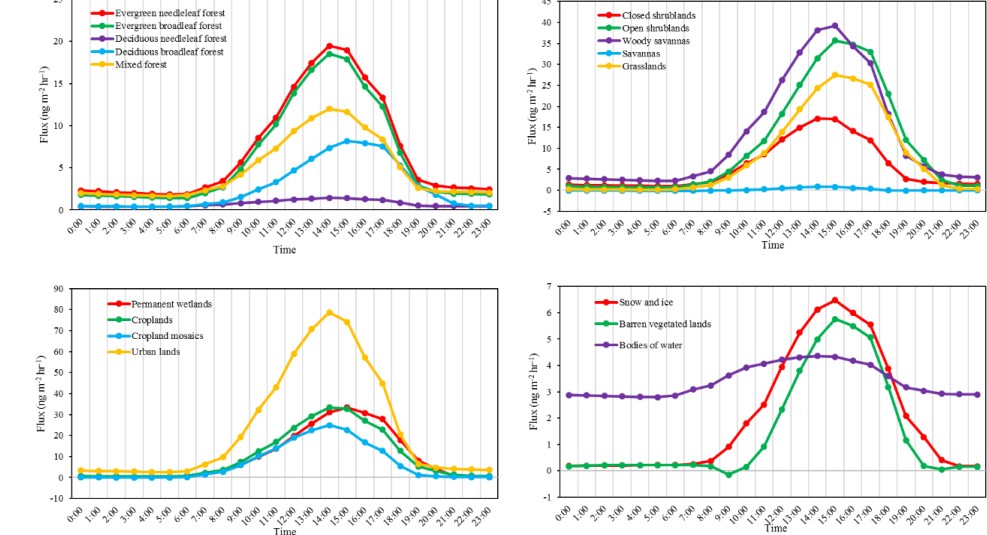



Figure 5. Diurnal variation of mean simulated exchange fluxes of Hg$^0$ over soil and water surfaces in the
model domain (UTC+8).





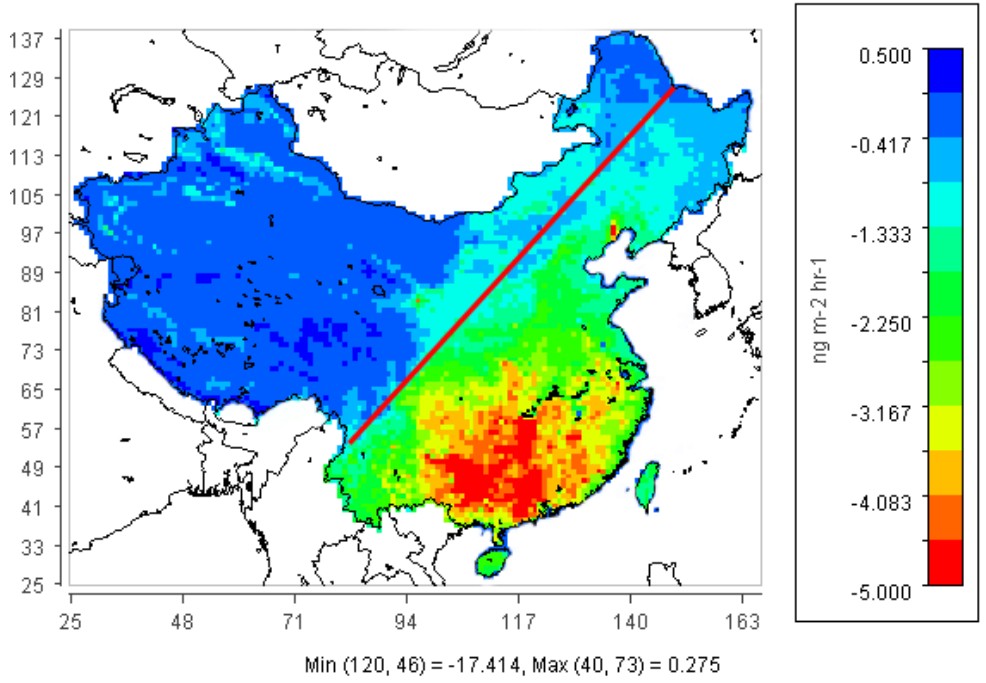


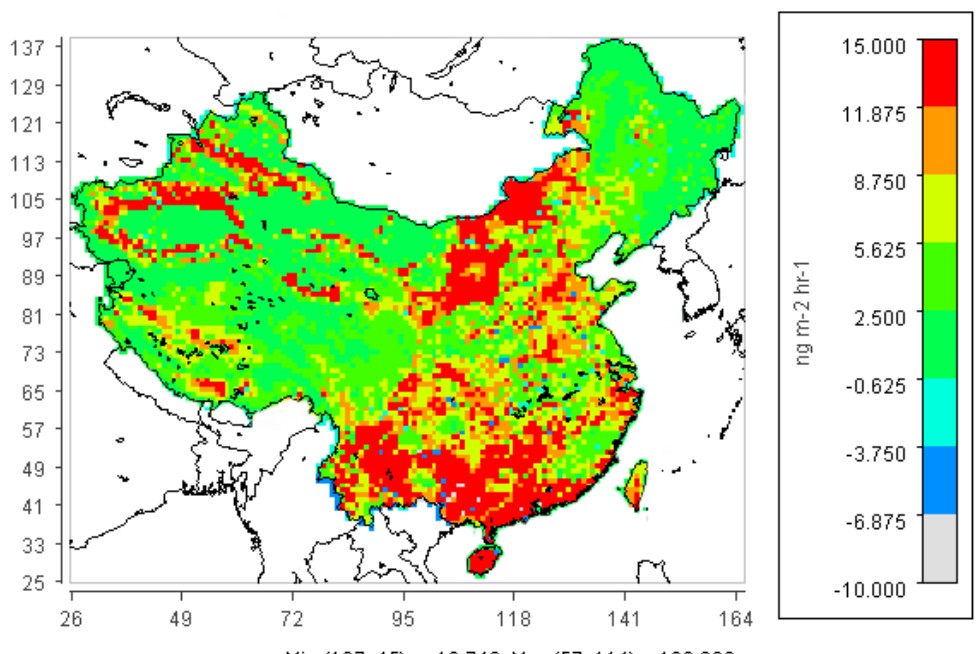


Figure 6. Simulated results of (1) mean annual air-foliage flux, and (2) mean annual air-soil flux in the
study domain.

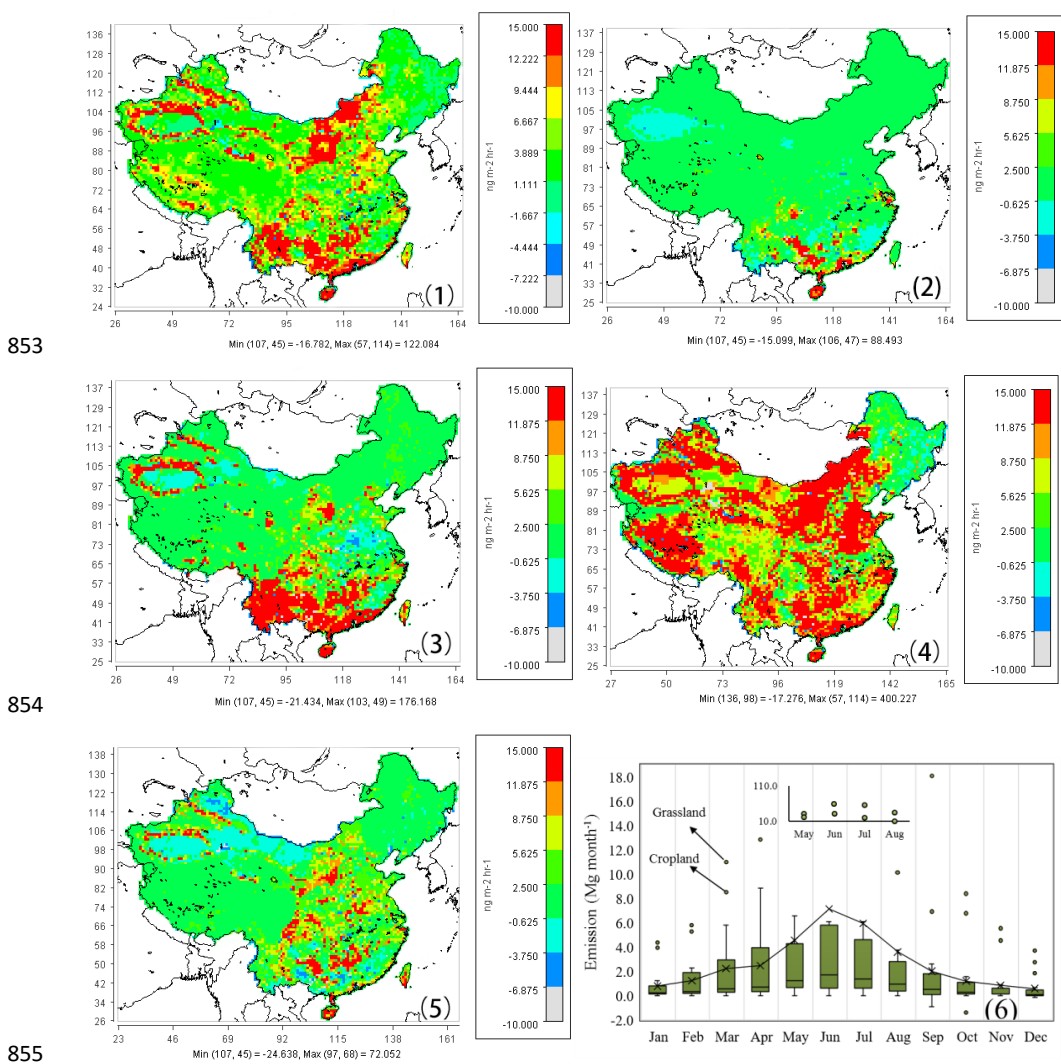


Figure 7. Model estimates of (1) annual mean Hg⁰ fluxes in the model domain; (2) seasonal mean Hg⁰ fluxes in winter, (3) seasonal mean Hg⁰ fluxes in spring, (4) seasonal mean Hg⁰ fluxes in summer, (4) seasonal mean Hg⁰ fluxes in autumn, and (6) monthly Hg⁰ fluxes in the grid cells (box and whisker chart showing maximum, 75th percentile, mean, median, 25th percentile, and minimum).






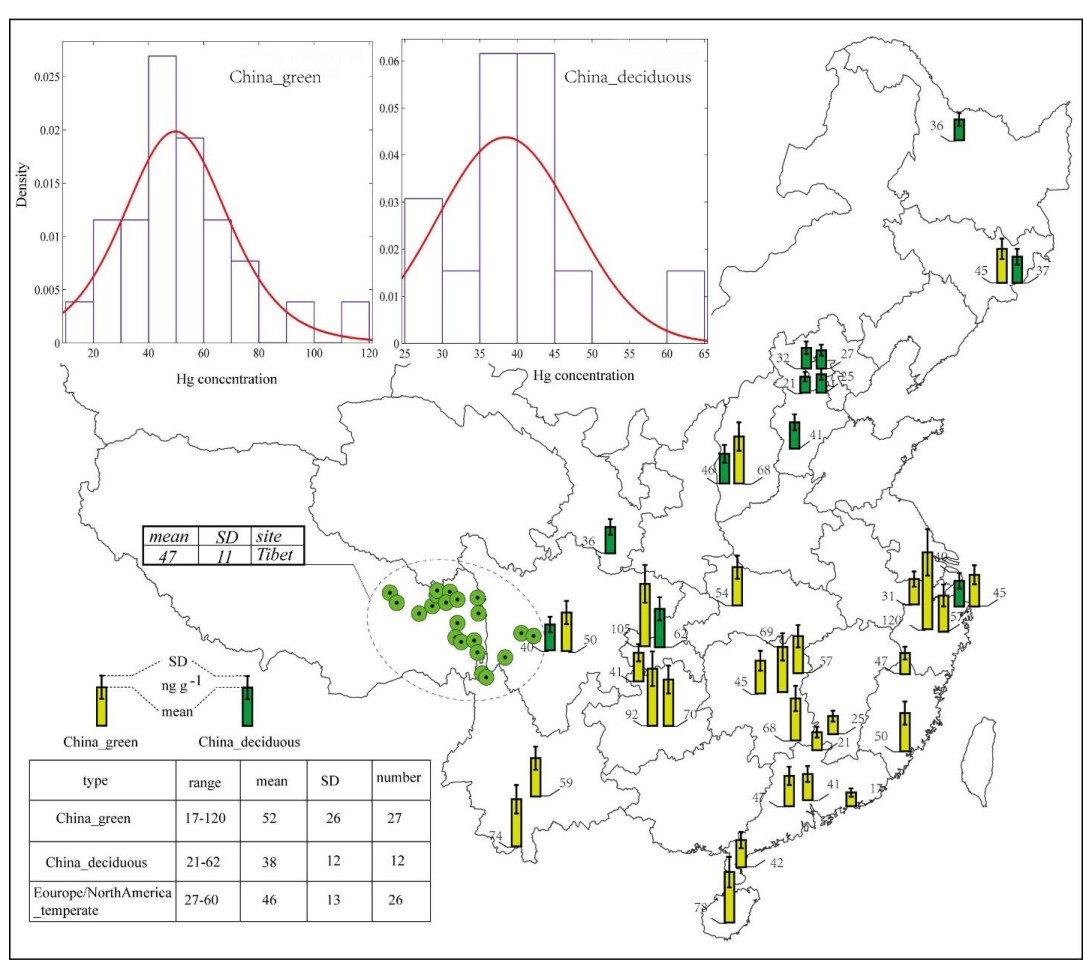


Figure 8. Database of Hg concentration in litterfall samples, China (Ma et al., 2015;Niu et al., 2011;Zhou et al., 2013;Fu et al., 2015;Wang et al., 2014a;Tang et al., 2015;Juillerat et al., 2012;Blackwell et al., 2014;Risch et al., 2012;Selvendiran et al., 2008). An unpublished dataset including 8 sites in China is described in details in the SI. The Hg concentrations in evergreen and deciduous forests have a t Location-Scale distribution ($\mu$=50.1, $\sigma$=19.3, F=6.6; and $\mu$=36.3, $\sigma$=3.6, F=1.4, respectively).

870



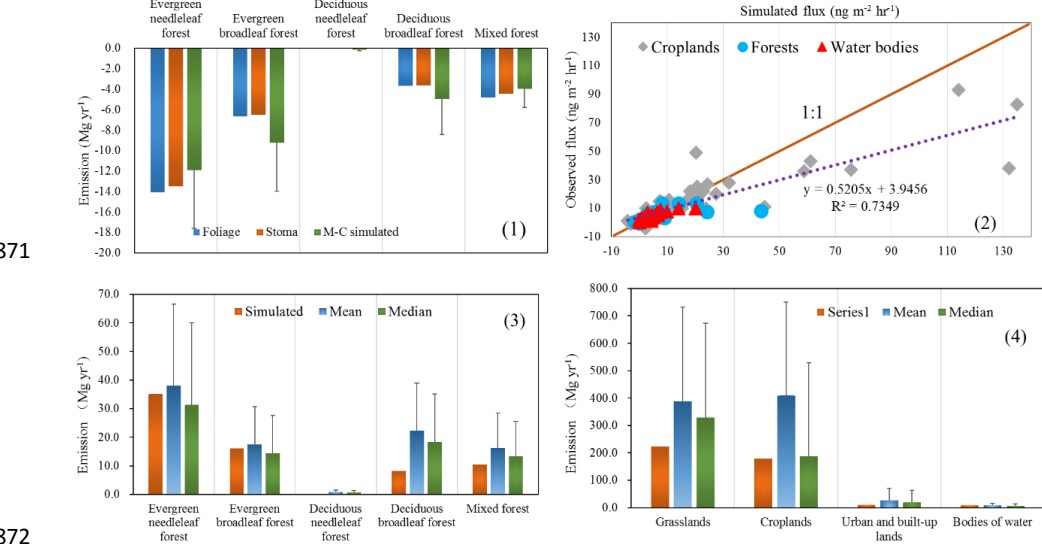

871

872

Figure 9. Model verification: (1) model estimates of $Hg^0$ uptake by foliage (which include the uptake by stoma less the re-emission and from cuticle) and by stoma, compared to the estimate (mean and 95% confidence interval) of $Hg^0$ uptake using Monte Carlo (M-C) simulation of the observational data; (2) scatter plot of the observed fluxes vs. simulated fluxes for different landuses (the flux observations are described in detailed in Table S2), (3) comparison between simulated exchange and measured exchange over soil under canopy, and (4) comparison between simulated exchange and measured exchange over grasslands, cropland and water surface. The mean and median of Figure 9.3 and 9.4 are based on the filed data from peer-review literatures (n=19 for forests; n=12 for grasslands; n=42 for croplands; n=51 for water bodies). Note that the exchange over deciduous forests in Figure 9.1 is small because of the small forest area.

883

884