# Peer review of "Emission-dominated gas exchange of elemental mercury vapor over natural surfaces in China 1 Xun Wang1,2, Che-Jen Lin1,3,4,\*, Wei Yuan1,2, Jonas Sommar1, Wei Zhu1, Xinbin Feng1,\* 2 3 4 1 State Key Laboratory of Environment"

_Atmospheric Chemistry and Physics, 2016_

## Referee Comment (RC1) · Anonymous Referee #1 · 7 Jun 2016

This paper argues the role and the large uncertainty of Hg emission from natural surfaces, very relevant topic in the study of Hg cycle. Particularly the authors analyze emission of elemental Hg0 from natural surfaces in China, developing a specific mechanistic model for estimate these. It is very accurate and well described the Section 2.1, which will be useful for further scientific developments. The conclusions reached are interesting, and the methods described in a clear way.

I found this paper interesting and I believe it should be published. However, I believe it should be slightly revised. In particular:

Abstract, lines 23-25: I believe should be better described the fluxes, in this form there is confusion (even in lines 318-320). Need of Table 3 to understand well the sign (up or down) of the flows. The sentence should be simplified

line 77: What is DOM?

line 331: Specify what it refers the percentage

line 336: The future projection is not clear. The sentence should be simplified.

line 835: Probably should be replaced "is" with "are"

figure 4: Set the y-axis in the interval [1,-11] in the two panels

In general the significant digits should be corrected (for example, line 250, tab1, etc)

Moreover, can the model (or the most relevant routines) to be made available in the supplementary information? It would be very useful for those who want to deepen or use these results

---

## Referee Comment (RC2) · Anonymous Referee #2 · 9 Jun 2016

This paper provides an updated model of estimating the exchange of gaseous elemental mercury between the atmosphere and underlying surfaces in China. This topic is relevant to the scope of ACP and of great importance as very high uncertainty still exists in its estimate and it is closely related to Hg input into the ecosystem in China and Hg export to downwind regions. In general, the manuscript is very well written. I suggest the acceptance of this paper after the authors addressing the following general and specific comments. The manuscript also needs a careful proofread to correct many grammatical errors. General Comments: 1. In this paper, the "natural" emission refers to the sum of the primary natural activities (i.e. geogenic) and the re-emission of legacy Hg stored in the terrestrial and water surfaces. However, in some other papers, the "natural" emission only refers to geogenic activities. This may cause some confusion. I suggest a clear definition of the "natural" emission be given in the introduction

section. 2. There is a logical question. It is mentioned in the paper that "the feedback of the air-surface exchange to the air concentration does not significantly modify the atmospheric Hg0 concentration" (Lines 226-229), and thus I think that a different air-surface exchange mechanism may not significantly affect the outflow of Hg from China. But the authors do argue in the paper that the effect may be large. 3. The updated soil Hg concentration dataset is based on the NMPRGS survey. What is the depth of soil sampling, 0-10 cm? 0-20 cm? It seems that most of HgII reduction occurs within the very top soil layer. What is the vertical profile of Hg concentration? 4. It may be beyond the scope of this specific paper, but I wonder is it possible to calculate an uncertainty range of natural Hg emission? Specific Comments: Lines 51-52: The trend of future anthropogenic Hg emission depends on many factors and the Minamata Convention does not necessarily lead to a decrease. Also, the reference De Simone et al. 2015 is not relevant here. Lines 195-197: Is the value of the parameter of soil interfaces compensation point (3 ng/m3) only based on the measurements at air-snow interface? What about other types of underlying surfaces? Lines 224-226: Are the CMAQ modeled atmospheric Hg0 concentrations validated against real world observations in China? Lines 248-251: Do the measurements in the surface forest floor have similar environmental conditions with the modeling? In addition, the average measured concentration of 4.1 ng/m3 seems to be very high, if we consider vegetation serves as a large sink. Lines 257: What does the 1-2 ng/m3 refer to? Is it the difference between soil and air Hg concentrations? Technical Corrections: Line 64: "may be appropriate"; Line 69: "estimate"; Line 96: "at a resolution of"; Line 98: "a recent database"; Line 199: "at a 36-km"; Line 286: "can improve"; Line 402: "are more comprehensive"; Line 481: "to earlier estimates"; Line 487: "also enhances"; Figure 9.4: missing legend;

---

## Referee Comment (RC3) · Anonymous Referee #3 · 5 Jul 2016

The paper proposes a new parametrization of surface fluxes of elemental mercury based on different pathways of reduction of reactive mercury. The model is applied to China, taking into account landuse patterns to estimate emissions of elemental mercury.

The main problem with the study is the minimal evaluation of the model and the lack of details concerning the sensitivity studies. The organization of the results section was puzzling: Sec 3.4 has the evaluation of the model, which I would have expected at the beginning of Section 3. I would have liked to see more details about the evaluation. Figure 8 shows only measurements and was difficult to figure out, it needs a better legend (eg. Units) and the information should be presented in such a way as to help evaluate the model results. Figure 9 contains the information for evaluating the model, but it is difficult to get a clear sense of model performance from this.

[Figure]

The sensitivity study seemed very limited in scope, with only a low and a high level. It seems there could be a more thorough way of doing this. Figure 2 was difficult to see – cross-sections would probably be preferable. For Figure 3, I was surprised at the magnitude of the changes (around 100 ng m-2 hr-1) when the fluxes listed in Table 3 are 1 to 2 orders of magnitude smaller.

Because the model is very specific in inputs, it seems to me that the model development part requires a very specific evaluation which is distinct from the application of the model on the national scale. The paper therefore seems to be a curious combination of 2 papers: one paper on model development and one on application of the model to a national scale. However, I think the paper would be acceptable with an expanded description of the model evaluation and an improved sensitivity analysis.

Specific comments:

Sec 2.1.1: It would be good to explain how the model differs from prior work in more detail.

Sec 3.1: It is preferable to talk about "evaluation" rather than "verification." Model evaluation seems to be in Sec 3.4. Sec 3.1 seems to be a comparison with other studies – a graphical representation may help some of the discussion.

Line 222: Putting uncertainty on the bounds of the ranges seemed like an odd thing to do. Isn't it enough to state the range?

---

## Author Comment (AC1) · 7 Aug 2016

This paper argues the role and the large uncertainty of Hg emission from natural surfaces, very relevant topic in the study of Hg cycle. Particularly the authors analyze emission of elemental Hg0 from natural surfaces in China, developing a specific mechanistic model for estimate these. It is very accurate and well described the Section 2.1, which will be useful for further scientific developments. The conclusions reached are interesting, and the methods described in a clear way. I found this paper interesting and I believe it should be published. However, I believe it should be slightly revised.

Response: We thank the reviewer for recognizing the significance of our work, and appreciate the reviewer's constructive comments, which have been addressed in our response to the specific comments below.

[Figure]

In particular: Abstract, lines 23-25: I believe should be better described the fluxes, in this form there is confusion (even in lines 318-320). Need of Table 3 to understand well the sign (up or down) of the flows. The sentence should be simplified.

Response: We agree and have simplified the sentence as "The net exchange of Hg0 between the atmosphere and natural surfaces of Mainland China is estimated to be 465.1 Mg yr-1, including 565.5 Mg yr-1 from soil surfaces, 9.0 Mg yr-1 from water body, and -100.4 Mg yr-1 from vegetation" in Line 23-25 and Line 328-329.

line 77: What is DOM?

Response: DOM stands for dissolved organic matter. It has been clarified in Line 75.

line 331: Specify what it refers the percentage

Response: The percentage refers to the fraction of forested area in China. The sentences has been revised as "From 2000 to 2013, the forested area in China increased from 14.0% to 21.6%" in Line 341.

line 336: The future projection is not clear. The sentence should be simplified.

Response: We thank for the reviewer for pointing this out. The sentence has been revised as "Given the forest coverage is projected to be 24 to 26% during 2030-2050 (FAO, 2014), the quantity of natural Hg emission in China would decrease by 9-10% compared to the estimated level of 2013" in Line 345-346.

line 835: Probably should be replaced "is" with "are"

Response: The verb has been corrected.

Figure 4: Set the y-axis in the interval [1,-11] in the two panels.

Response: We have reset the interval for y-axis in Figure 4.

In general the significant digits should be corrected (for example, line 250, tab1, etc).

Response: We have make the number of significant figures consistent with the estimate throughout the manuscript.

Moreover, can the model (or the most relevant routines) to be made available in the supplementary information? It would be very useful for those who want to deepen or use these results.

Response: That is a very good suggestion. We also plan to publish the code of this model. Given the complexity of this model, we are planning to write a user guide, and will publish these in our website later (http://www.973hg.org/Default.aspx).
* * *

---

## Author Comment (AC2) · 7 Aug 2016

This paper provides an updated model of estimating the exchange of gaseous elemental mercury between the atmosphere and underlying surfaces in China. This topic is relevant to the scope of ACP and of great importance as very high uncertainty still exists in its estimate and it is closely related to Hg input into the ecosystem in China and Hg export to downwind regions. In general, the manuscript is very well written. I suggest the acceptance of this paper after the authors addressing the following general and specific comments. The manuscript also needs a careful proofread to correct many grammatical errors.

Response: We thank the reviewer for recognizing the contribution of our work and for the constructive comments, which have been addressed in our response below and in

[Figure]

the revised manuscript. We also appreciate the reviewer for taking the time to provide the editorial comment.

General Comments:

1. In this paper, the "natural" emission refers to the sum of the primary natural activities (i.e. geogenic) and the re-emission of legacy Hg stored in the terrestrial and water surfaces. However, in some other papers, the "natural" emission only refers to geogenic activities. This may cause some confusion. I suggest a clear definition of the "natural" emission be given in the introduction section.

Response: We totally agree with the reviewer on this comment and have clarified this in Line 41-44: The inventories of Hg emission include the emission from anthropogenic sources, and the so-called "natural" emission. In this study, the term "natural emission" refers to the sum of the primary natural release (i.e., from geogenic activities) and the re-emission of legacy Hg stored in the terrestrial and water surfaces, because the geogenic release and re-emission cannot be separated analytically.

2. There is a logical question. It is mentioned in the paper that "the feedback of the air-surface exchange to the air concentration does not significantly modify the atmospheric Hg0 concentration" (Lines 226-229), and thus I think that a different air-surface exchange mechanism may not significantly affect the outflow of Hg from China. But the authors do argue in the paper that the effect may be large.

Response: We understand the reviewer's point and wat to clarify here. In short, the total mass of release is significant but the concentration forced by the release is insignificant. Outflow is a mass budget and therefore will be affected by the quantity of Hg natural emission. Given the $\sim$ 1 km mixing layer for each grid cell, the feedback of the air-surface exchange to the air Hg concentration is 0.01-0.1 ng m-3. However, if we consider the area of our domain (9,600,000,000,000 m2), and the several days of Hg0 residence time in our domain, the Hg output during this period can be up to several tons (Mg).

[Figure]

3. The updated soil Hg concentration dataset is based on the NMPRGS survey. What is the depth of soil sampling, 0-10 cm? 0-20 cm? It seems that most of HgII reduction occurs within the very top soil layer. What is the vertical profile of Hg concentration?

Response: The surface soil depth is 0-20 cm. Though the mean Hg concentration in 0-20 cm soil profile could somewhat underestimates Hg concentration in the top soil layer, the dataset is the best available one describing the soil Hg concentration in China. We have made this clear in the revised manuscript (Line 223-226): Datasets of Hg concentration in the top soil layer (e.g., 0-5 cm depth) are recommended for simulations when they become available.

4. It may be beyond the scope of this specific paper, but I wonder is it possible to calculate an uncertainty range of natural Hg emission?

Response: That is a very good suggestion. Mathematically, it is possible to calculate the uncertainty for a mechanistic model. However, it is beyond the scope of this study because the uncertainty is controlled by the input parameters/datasets, and the uncertainty for certain dataset has not been systematically quantified, the uncertainty range for the Hg concentration in surface soil. With that said, on the basis of the mode verification results, the estimates in this study is generally in agreement with the field observations.

Specific Comments:

Lines 51-52: The trend of future anthropogenic Hg emission depends on many factors and the Minamata Convention does not necessarily lead to a decrease. Also, the reference De Simone et al. 2015 is not relevant here.

Response: We agree that future anthropogenic Hg emission depends on many factors. We have replaced this reference by Pacyna et al. (2016) and revised the sentence as "In light of the control of anthropogenic Hg emission by the legally binding Minamata Convention, a better quantification of natural Hg emission is critical in evaluating the

effectiveness of policy actions (Selin, 2009;Pirrone et al., 2010;Song et al., 2015)" in Line 51-54.

Lines 195-197: Is the value of the parameter of soil interfaces compensation point (3 ng/m3) only based on the measurements at air-snow interface? What about other types of underlying surfaces?

Response: Yes, the measured compensation point (3 ng/m3) is only for the surface covered by snow. The compensation point for other types of soil surface is calculated based on Equations 1-18. We have clarified in Line 194-195 : The $\chi\_g$ for the air-snow interface is assumed to be 3 ng m-3 based on field measurements at air-snow interface.

Lines 224-226: Are the CMAQ modeled atmospheric Hg0 concentrations validated against real world observations in China?

Response: Yes, the CMAQ model results have been verified. More details of the model simulations are presented in Lin et al. (2010).

Lines 248-251: Do the measurements in the surface forest floor have similar environmental conditions with the modeling? In addition, the average measured concentration of 4.1 ng/m3 seems to be very high, if we consider vegetation serves as a large sink.

Response: Yes, the measurements in the surface forest floor have similar environmental conditions to the modeling, and we have clarified this in Line 822-823. We want to clarify here the case in which the mean measured concentration is 4.1 ng/m3. This is the Hg0 vapor concentration in soil porous media, not the concentration in ambient air.

Lines 257: What does the 1-2 ng/m3 refer to? Is it the difference between soil and air Hg concentrations? Response: It refers to the atmospheric Hg concentration at background site. To clarify it, we have revised the sentence as "Hg0 soil gas concentrations are typically lower than the 1-2 ng m-3 atmospheric Hg concentration in background forest ecosystems at night." (Line 259-260)

Technical Corrections: Line 64: "may be appropriate"; Response: The wording has been revised as suggested. Line 69: "estimate"; Response: The wording has been revised as suggested. Line 96: "at a resolution of"; Response: The wording has been revised as suggested. Line 98: "a recent database"; Response: The wording has been revised as suggested. Line 199: "at a 36-km"; Response: The wording has been revised as suggested. Line 286: "can improve"; Response: The wording has been revised as suggested. Line 402: "are more comprehensive"; Response: The wording has been revised as suggested. Line 481: "to earlier estimates"; Response: The wording has been revised as suggested. Line 487: "also enhances"; Response: The wording has been revised as suggested. Figure 9.4: missing legend; Response: The missing legend has been added in Figure 8.4.

---

## Author Comment (AC3) · 7 Aug 2016

The paper proposes a new parametrization of surface fluxes of elemental mercury based on different pathways of reduction of reactive mercury. The model is applied to China, taking into account landuse patterns to estimate emissions of elemental mercury.

Response: We thank the reviewer for recognizing the new contribution of our work, and appreciate the reviewer's constructive comments.

The main problem with the study is the minimal evaluation of the model and the lack of details concerning the sensitivity studies. The organization of the results section was puzzling: Sec 3.4 has the evaluation of the model, which I would have expected at the beginning of Section 3. I would have liked to see more details about the evaluation.

[Figure]

Response: We thank reviewer for the suggestion and have provided additional data in Figures S8-S9 to address the reviewer's concern regarding model evaluation. We respectfully disagree regarding the organization of the section describing the results. Compared to earlier air-surface models (Bash, 2010;Wang et al., 2014), we built a new scheme for air-soil flux exchange. We think the evaluation for this scheme is necessary before running the air-surface exchange model and choose to present the evaluation results of individual model components in section 3.1, and show the verification results after multiple model components are integrated in section 3.4.

Figure 8 shows only measurements and was difficult to figure out, it needs a better legend (eg. Units) and the information should be presented in such a way as to help evaluate the model results. Figure 9 contains the information for evaluating the model, but it is difficult to get a clear sense of model performance from this.

Response: We have provided additional data in Figures S8-S9 to address such concerns. Figure S8 and S9 display the comparison for the air-foliage flux exchange by the model and field data of Hg deposition through litterfall. We have revised this section in Line 441-452 as: "Figures S8, S9 and 8.1 compare the model estimates to the mean and variability level predicted by Monte Carlo simulation using field data. The annual Hg uptake simulated by the bidirectional exchange model is not significantly different from the field observations ($p > 0.05$, t-test) and the spatial patters are similar (Figure S8) in coniferous forest ecosystems, demonstrating the model capability for simulating the air-foliage flux. However, the bidirectional exchange model did not capture the spatial distribution of air-foliage flux in broadleaf forest ecosystems (particularly in evergreen broadleaf forest, Figure S9). One possible explanation is that the resistance terms obtained from temperate/boreal forests (Zhang et al., 2012b) may not appropriately represent the value in evergreen broadleaf forests. Filed measurements suggests that the leaf stomatal conductance of broadleaf is usually higher than the value of needleleaf (Wang et al., 2015;Ishida et al., 2006;Sobrado, 1991;Eamus, 1999), leading to a larger air-foliage $Hg^0$ exchange (Graydon et al., 2006). Further studies on the

Hg transport and chemical reactions at the air-foliage interface in evergreen broadleaf forests will help constrain the model. "

The sensitivity study seemed very limited in scope, with only a low and a high level. It seems there could be a more thorough way of doing this.

Response: We thank the reviewer for pointing this out and want to clarify here. The 2-level factorial design of experiments is meant to gauge the extreme variation of flux caused by the possible range of all parameters. This method is statistically robust, and therefore the synergistic and antagonistic interactions among model parameters can be estimated with indications of statistical significance. In reality, since the actual variation of the parameters is much smaller than the possible range, the flux change will also be much milder. To illustrate this, we run the model using the center value (i.e., showing the model results by running the model at half of the experimental level). By using the center values of soil Hg content, LAI, soil bulk density, solar radiation and soil temperature in Table 2 (close to the environmental parameters in a typical forest ecosystem), the air-soil flux is 4.5 ng m-2 hr-1. Such flux is close to the measures fluxes (0.5-9.3 ng m-2 hr-1) in forest ecosystems of China (Fu et al., 2012;Fu et al., 2008). We have supplement the above discussion in the Line 267-287.

Figure 2 was difficult to see – cross-sections would probably be preferable.

Response: We have added the cross-sections for a better presentation.

For Figure 3, I was surprised at the magnitude of the changes (around 100 ng m-2 hr-1) when the fluxes listed in Table 3 are 1 to 2 orders of magnitude smaller.

Response: We carefully checked the process of the sensitivity analysis, and found that we omitted the parameter of ratio of UV radiation over total radiation (Table 1). In revised Figure 3, the flux change is about from 20-30 ng m-2 h-1. Filed measurements suggest the combined effects of soil Hg content (from 60 to 590 ng g-1) and soil temperature (from 5 to 30 ËŽC) enhance the flux by ∼40 ng m-2 hr-1 (Fu et al., 2012;Fu

et al., 2008). (Line 274-276).

Because the model is very specific in inputs, it seems to me that the model development part requires a very specific evaluation which is distinct from the application of the model on the national scale. The paper therefore seems to be a curious combination of 2 papers: one paper on model development and one on application of the model to a national scale. However, I think the paper would be acceptable with an expanded description of the model evaluation and an improved sensitivity analysis.

Response: We appreciate the reviewer's suggestion and want to clarify here. We agree that a specific evaluation for the model development is necessary and this have been thoroughly discussed in section 3.1 (Line 238-296). It is our view that a comprehensive paper describing the model development and application serve readers better.

Specific comments:

Sec 2.1.1: It would be good to explain how the model differs from prior work in more detail.

Response: We agree with the reviewer on the comment and have added a section for this purpose in Line 111-115: "Compared to the earlier mechanistic schemes (Wang et al., 2014;Bash, 2010;Scholtz et al., 2003;Zhang et al., 2012), this model (1) builds a new scheme for the air-soil flux based on the reduction pathways of reactive Hg in soil identified in the literature, (2) develops a scheme for the Hg flux exchange in rice paddy, which is an important landuse feature in China, and (3) updates the scheme for the air-snow interface and chemical parameters for air-foliage flux (Table 1)."

Sec 3.1: It is preferable to talk about "evaluation" rather than "verification." Model evaluation seems to be in Sec 3.4. Sec 3.1 seems to be a comparison with other studies – a graphical representation may help some of the discussion.

Response: We thank the reviewer for the constructive suggestion. We have used the evaluation of the air-soil flux scheme in section 3.1 of revised manuscript.

Line 222: Putting uncertainty on the bounds of the ranges seemed like an odd thing to do. Isn't it enough to state the range? Response: Thanks for constructive suggestions. This sentence has been revised as "The soil Hg content in 0-20 cm surface soil varies with landuse types, containing mean concentrations of 119~211, 61~197, 80~82, 80~82 and 31~162 of Hg for forest ecosystems".

References Bash, J. O.: Description and initial simulation of a dynamic bidirectional air-surface exchange model for mercury in Community Multiscale Air Quality (CMAQ) model, J Geophys Res-Atmos, 115, 2010. FAO: State of the World's Forests 2014, FOOD AND AGRICULTURE ORGANIZATION OF THE UNITED NATIONS, Rome, 93-100, 2014. Fu, X. W., Feng, X. B., and Wang, S. F.: Exchange fluxes of Hg between surfaces and atmosphere in the eastern flank of Mount Gongga, Sichuan province, southwestern China, Journal of Geophysical Research-Atmospheres, 113, 2008. Fu, X. W., Feng, X. B., Zhang, H., Yu, B., and Chen, L. G.: Mercury emissions from natural surfaces highly impacted by human activities in Guangzhou province, South China, Atmos Environ, 54, 185-193, 2012. Lin, C. J., Pan, L., Streets, D. G., Shetty, S. K., Jang, C., Feng, X., Chu, H. W., and Ho, T. C.: Estimating mercury emission outflow from East Asia using CMAQ-Hg, Atmos Chem Phys, 10, 1853-1864, 2010. Pacyna, J. M., Travnikov, O., De Simone, F., Hedgecock, I. M., Sundseth, K., Pacyna, E. G., Steenhuisen, F., Pirrone, N., Munthe, J., and Kindbom, K.: Current and future levels of mercury atmospheric pollution on global scale, Atmos. Chem. Phys. Discuss., 2016, 1-35, 10.5194/acp-2016-370, 2016. Pirrone, N., Cinnirella, S., Feng, X., Finkelman, R. B., Friedli, H. R., Leaner, J., Mason, R., Mukherjee, A. B., Stracher, G. B., Streets, D. G., and Telmer, K.: Global mercury emissions to the atmosphere from anthropogenic and natural sources, Atmos Chem Phys, 10, 5951-5964, 2010. Scholtz, M. T., Van Heyst, B. J., and Schroeder, W.: Modelling of mercury emissions from background soils, Science of the Total Environment, 304, 185-207, 10.1016/s0048-9697(02)00568-5, 2003. Selin, N. E.: Global Biogeochemical Cycling of Mercury: A Review, Annu Rev Env Resour, 34, 43-63, 2009. Song, S., Selin, N. E., Soerensen, A. L., Angot, H., Artz, R., Brooks, S., Brunke, E. G., Conley, G., Dommergue, A., Ebinghaus, R., Holsen, T.

M., Jaffe, D. A., Kang, S., Kelley, P., Luke, W. T., Magand, O., Marumoto, K., Pfaffhuber, K. A., Ren, X., Sheu, G. R., Slemr, F., Warneke, T., Weigelt, A., Weiss-Penzias, P., Wip, D. C., and Zhang, Q.: Top-down constraints on atmospheric mercury emissions and implications for global biogeochemical cycling, Atmos Chem Phys, 15, 7103-7125, 2015. Wang, X., Lin, C. J., and Feng, X.: Sensitivity analysis of an updated bidirectional air-surface exchange model for elemental mercury vapor, Atmos Chem Phys, 14, 6273-6287, 2014. Zhang, L., Blanchard, P., Johnson, D., Dastoor, A., Ryzhkov, A., Lin, C. J., Vijayaraghavan, K., Gay, D., Holsen, T. M., Huang, J., Graydon, J. A., St Louis, V. L., Castro, M. S., Miller, E. K., Marsik, F., Lu, J., Poissant, L., Pilote, M., and Zhang, K. M.: Assessment of modeled mercury dry deposition over the Great Lakes region, Environ Pollut, 161, 272-283, 2012.
* * *